# Femtosecond laser synthesis of multiscale high-entropy alloys/graphene composites for high-performance Joule heating

Lingxiao Wang [1], Kai Yin [1,2] ✉, Jianqiang Xiao[1], Xinghao Song[1], Jiaqing Pei[1], Jun He [1] & Ji-An Duan[2]

High-entropy alloy nanoparticles (HEA-NPs) have garnered significant interest across diverse fields. However, thus far, research on their applications has predominantly focused on electrocatalysis. Expanding the applications of HEA-NPs beyond current fields is timely and desirable but remains a challenge. Here, we demonstrate the successful femtosecond laser synthesis of HEA-NPs on the laser-induced graphene (LIG) for realizing high-performance Joule heating applications. This prepared composites (HEAs/LIG) exhibits exceptional electrothermal conversion ability with efficiency up to ~285.4 °C cm² W⁻¹. Furthermore, the HEAs/LIG also shows high broadband infrared emissivity of ~0.98 across the wavelength range from 2.5 to 20 μm. Finally, we present the applications of HEAs/LIG as an efficient Joule heater, which consumes ~49.1% less energy compared to conventional electrical heaters in winter. This work expands the application of HEA-NPs into the Joule heating field, and underlining the importance of further development in efficient energy utilization technology.

Since the concept of high-entropy alloys (HEAs) was first proposed in 2004, the HEAs have emerged as a frontier in materials science due to their unique properties[1–5]. The multicomponent solid-solution alloys results in high configurational entropy and synergism among multiple components, which endows HEAs with remarkable mechanical[6], thermal[7], and chemical properties[8,9]. Driven by advances in nanotechnology, the preparation of HEA nanoparticles (HEA-NPs) has become convenient. HEA-NPs exhibit unique properties, including elevated specific surface areas, compositional tunability, and pronounced lattice distortions. Tunable electronic structures and abundant active sites render HEA-NPs particularly attractive to be used as a versatile platform[10–12]. Consequently, HEA-NPs play an essential role in numerous applications including catalysis[8,13], batteries[14], solar evaporation[15,16], and electromagnetic shielding[17], among many others (Supplementary Fig. 1). The combination of these structural and functional advantages

highlights their potential and nanoscale derivatives in next-generation material applications.

Joule heating applications impose a demanding combination of high broadband infrared (IR) emissivity and high conductivity. To date, most research of HEA-NPs has concentrated on their catalytic activity, with comparatively limited exploration of other functional applications. Leveraging the inherent metallic characteristics, HEA-NPs typically demonstrate good electrical conductivity[18,19]. For the optical properties, recent studies have indicated that HEA-NPs exhibit strong absorption through d-d interband transitions in the visible range[20], while high-entropy oxides (HEOs) can demonstrate both high absorption and IR emissivity[21–24]. These findings reveal the potential for coupling the conductivity of HEA-NPs with enhanced IR radiative performance, also establishing the foundation for their Joule heating application. However, a high-performance Joule heater based on HEA-NPs is timely and desirable yet difficult to realize due to the demanding

[1]Hunan Key Laboratory of Nanophotonics and Devices, School of Physics, Central South University, Changsha 410083, China. [2]State Key Laboratory of Precision Manufacturing for Extreme Service Performance, College of Mechanical and Electrical Engineering, Central South University, Changsha 410083, China. ✉e-mail: kaiyin@csu.edu.cn

higher requirements for HEA-NPs emissivity. Thus, it is attractive to develop an effective approach to synthesize multifunctional HEA-NPs for high-performance Joule heating.

Herein, a solid-phase synthesis method based on femtosecond laser processing is proposed, enabling the successful synthesis of HEA-NPs on laser-induced graphene (LIG). Using LIG as the substrate, the high peak temperature and heating/cooling rates drive the formation of HEA-NPs within nanosecond timescale. Density functional theory (DFT) calculations verify that the mechanism of electrical resistance reduction originates from the HEA-NPs loading and deoxidation renovating effect on LIG. Notably, the multiscale composite material (HEAs/LIG) can serve as Joule heating material, achieving a heater energy efficiency of ~285.4 °C cm² W⁻¹. Besides exceptional electro-thermal performance, the HEAs/LIG also exhibits an excellent IR emissivity of ~0.98 across a broad wavelength range of 2.5–20 μm. Therefore, the HEAs/LIG was ~49.1% more energy efficient when used as an electric heater in winter compared to a commercial electric heater (CEH). These findings highlight the strong competitiveness of HEAs-based multifunctional materials for developing high-performance Joule heaters.

## Results

### Femtosecond laser synthesis

The synthesis process of HEAs/LIG by femtosecond laser is illustrated (Fig. 1a and Supplementary Fig. 2). Based on the suitable mixing enthalpy and atomic radius, Fe, Co, Ni, Cr, Mn, and Ru were chosen as the metallic precursors (Supplementary Tables 1 and 2). According to the calculation, the mixing entropy of FeCoNiCrMnRu HEA-NPs could jump to 1.79 R (Supplementary Note 1). We also prepared low-entropy (FeCo) and medium-entropy (FeCoNiCr) that were named as LEAs/LIG and MEAs/LIG, respectively. For the HEA-NPs system, femtosecond laser can break through the thermodynamic energy barrier and pre-pare FeCoNiCrMnRu HEA-NPs with uniform element distribution (Supplementary Fig. 3). The pristine LIG (P-LIG) substrate exhibited superwetting behavior to the mixed precursor solution, making homogeneous loading of metal ions (Supplementary Figs. 4, 5). The femtosecond laser (~350 fs) was employed in ambient air to prepare HEA-NPs (Supplementary Fig. 6). In this study, P-LIG was selected as the substrate because its super-black offers a high laser absorption of ~98% (Supplementary Figs. 7, 8). The resulting superior photothermal conversion efficiency, coupled with the low specific heat capacity and

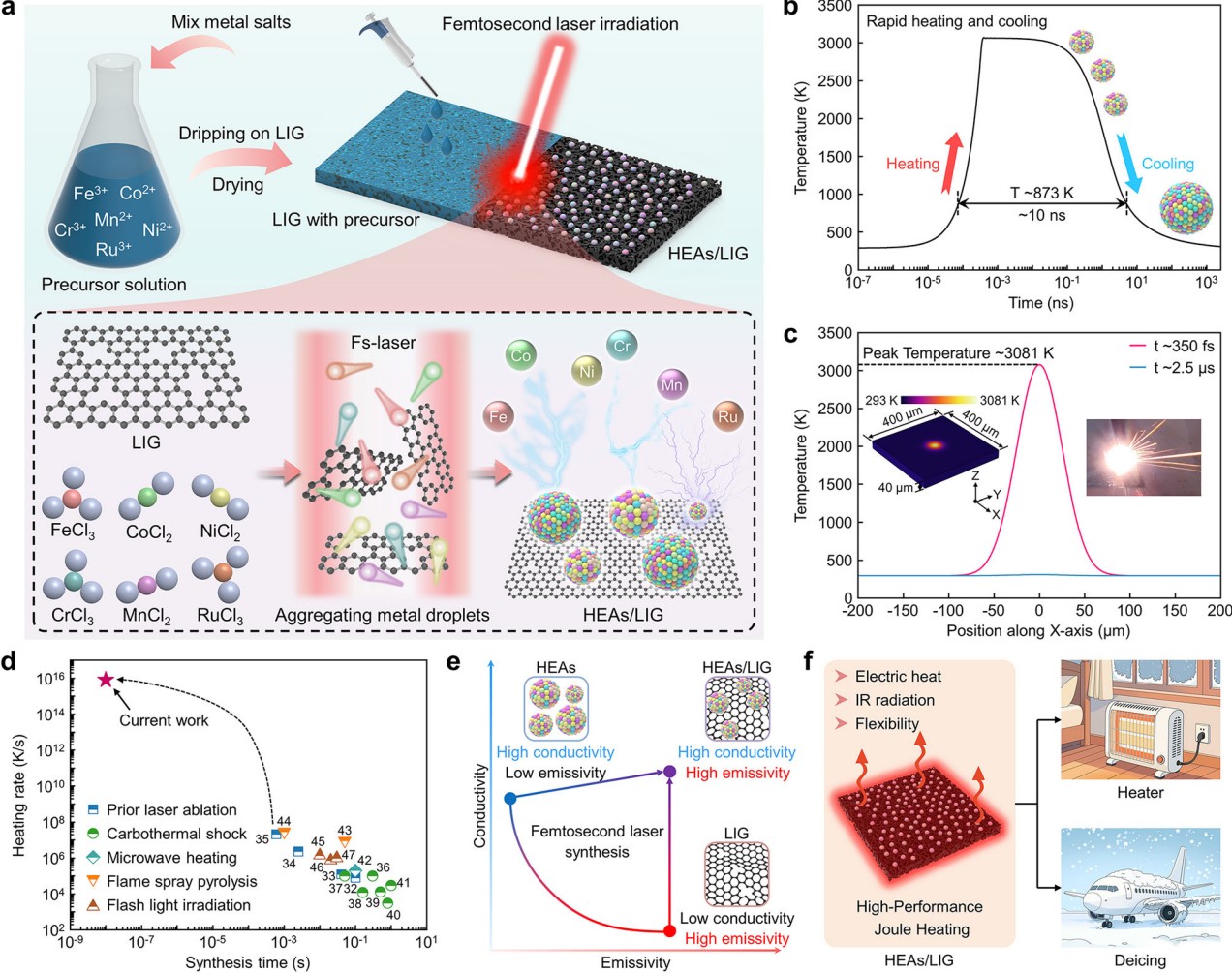

**Fig. 1 | Femtosecond laser synthesis process and application scenarios of HEAs/ LIG. a** Schematic representation and microscopic illustration for the synthesis of HEAs/LIG by femtosecond laser irradiation. **b** Simulated temperature-time curve of P-LIG under femtosecond laser irradiation. **c** Temperature field simulation of PI tape irradiated with a femtosecond laser pulse. The inset is the optical photo of femtosecond laser irradiating the P-LIG surface. **d** Comparison of the heating rate and synthesis time of femtosecond laser prepared HEA-NPs reported in the litera-ture (see Supplementary Table 4 for more details). **e** Schematic illustration of the design of high-performance Joule heating materials enabled by HEA-NPs and LIG. **f** Schematic illustration for the properties and application scenarios HEAs/LIG. Source data are provided as a Source Data file.

high IR emissivity, accelerates both the heating and cooling processes (Supplementary Fig. 9). Moreover, under the transient high-temperature conditions generated by femtosecond laser, the P-LIG layer can act as a micro-reactor that effectively blocks the binding of oxygen and metals, promoting the formation of single-phase HEA-NPs[25]. The temperature variation was simulated using COMSOL simulations (Fig. 1b and Supplementary Note 2). The simulated temperature-time curve on the surface demonstrated a heating time of 350 fs (one femtosecond laser pulse) to reach up to ~3081 K (Fig. 1c) with a heating rate of ~$8.0 \times 10^{15}$ K s$^{-1}$. Obviously, the high temperature far exceeded the pyrolysis temperatures of all metal salts, causing all precursors to decompose simultaneously (Supplementary Table 3). Subsequently, rapid alloy phase fixation was achieved with a cooling rate of ~$5.6 \times 10^{11}$ K s$^{-1}$ to avoid phase separation. The highly localized femtosecond laser pulse could facilitate rapid cooling and help preserve the single-phase random atomic configuration to promote the formation of uniform HEA solid solutions[3,9,10]. Additionally, the transient high temperature accelerated the nucleation process, restrict further particle growth, and form smaller particles (Supplementary Fig. 10 and Note 3). According to the simulation, the duration within a single femtosecond laser-induced thermal pulse that exceeds the pyrolysis temperatures of all metallic salts, which corresponds to the synthesis time of HEA-NPs, is ~10 ns. In addition, the high temperature only acted on the surface and did not damage the P-LIG substrate (Supplementary Fig. 11). It is noted that the classical heat diffusion model employed here provides an estimate of the extreme thermal cycle but does not capture the initial electron-lattice non-equilibrium. The predicted temperatures and heating/cooling rates should be interpreted as the qualitative conditions of the extreme conditions that facilitate rapid melting and solidification, consistent with the formation of HEA-NPs. While the exact numerical value is determined by the model, the phenomenon of rapid heating/cooling and its consequence is physically reasonable.

Femtosecond laser synthesis presents an effective route for the fabrication of HEA-NPs. This method leverages energy deposition on a femtosecond timescale, enabling the high heating rate and rapid synthesis durations[26–31]. Compared to previously reported methods for synthesizing HEA-NPs, such as prior laser ablation[32–35], carbothermal shock[36–41], microwave heating[42], flame spray pyrolysis[43,44], and flash irradiation[45–47], femtosecond laser has made progress in both heating rate and synthesis time (Fig. 1d and Supplementary Table 4). Furthermore, our method realized convenient synthesis of HEA-NPs, benefiting from non-contact processing, precise pattern control, and tunable parameters (Supplementary Table 5)[48–50]. Notably, compared with other research on the laser heating for HEAs, we successfully achieved the synthesis of multiscale HEAs/LIG and explored the electrical and IR radiation properties, significantly broadening the application of HEA-based materials. The femtosecond laser demonstrates obvious advantages over nanosecond and continuous wave (CW) lasers for the synthesis of HEA-NPs (Supplementary Table 6). Femtosecond laser synthesis enables the formation of significantly smaller HEA-NPs compared to those typically obtained with CW laser[51]. This is a direct consequence of the high heating and cooling rates, which rapidly freeze the nucleated liquid droplets, effectively suppressing coalescence that leads to coarse and irregular particles under the slower thermal cycles of CW laser. A key practical advantage is the ability to synthesize HEA-NPs directly in ambient atmosphere without requiring liquid environments or chemical reductants. Many nanosecond laser methods or other techniques for producing non-oxide HEA-NPs necessitate such controlled conditions[32–35,52–54]. The femtosecond laser achieves this through non-equilibrium energy deposition, while the short interaction time minimizes oxidation of HEA-NPs despite being performed in air. Our strategy for achieving high-performance Joule heating incorporated utilizing the high conductivity of HEA-NPs and high emissivity of LIG (Fig. 1e). Stable anchoring of HEA-NPs onto LIG

synergistically enhanced the electrothermal conversion performance while retaining the high IR emissivity of LIG substrate. Owing to its excellent Joule heating property, the HEAs/LIG exhibited promising potential for diverse heating applications, such as heaters and de-icing/snow-melting systems (Fig. 1f). Based on the concentration of the FeCoNiCrMnRu precursor solution (0.1 M), the used volume (160 μL) and the coated area (20 mm × 20 mm), we can estimate the mass loading of the FeCoNiCrMnRu HEA-NPs on the LIG to be ~1.5 mg cm$^{-2}$. Therefore, even if Ru was used with considering its cost and supply, its contribution to device-level cost is limited. However, it is also desirable to investigate a wider range of low-cost and abundant elements, such as Cu, Zn, Mo, and Sn, to either replace Ru or further increase the configurational entropy, which is anticipated to correspondingly enhance the electrothermal performance of the HEAs/LIG composite. We also fabricated a HEAs/LIG sample with an active area of 40 mm × 40 mm, and the infrared thermal image of this sample under an applied voltage revealed a highly uniform temperature distribution (Supplementary Fig. 12). This excellent uniformity is achieved by our femtosecond laser direct writing process, which involves point-by-point material transformation followed by a continuous line-scanning strategy. Laser systems with large-field lenses and higher-power lasers will be available for industrial-scale processing. We also evaluated the batch-to-batch reproducibility by measuring the sheet resistance at multiple locations across five independent HEAs/LIG samples fabricated over a span of eight month (Supplementary Fig. 13). The results show a very narrow distribution in sheet resistance (Supplementary Fig. 14). This remarkably low variation displayed the exceptional batch-to-batch reproducibility of our method. The processing time for our method preparing HEAs/LIG samples is directly proportional to the area. This represents a highly efficient and single-step process that simultaneously synthesizes the HEAs/LIG in ambient air. The most promising route for industrial-scale throughput is parallelization, which uses beam shaping with a spatial light modulator (SLM) to create multiple focused spots or a homogeneous flat-top beam, allowing for the parallel writing of multiple lines or large areas in a single pass.

## Structural properties and surface chemistry

The HEAs/LIG also exhibited excellent flexibility (Fig. 2a). The first femtosecond laser applied to the PI substrate surface produced fibrous porous micro/nanostructures, which was beneficial for uniformly loading metal precursors (Supplementary Figs. 15, 16). Following the second femtosecond laser, scanning electron microscopy (SEM) images revealed the formation of abundant irregular micrometer-scale protuberances covered with reticulated nanostructures on the HEAs/LIG surface (Fig. 2b and Supplementary Fig. 17). Experimental results demonstrated that as the laser scanning speed increased, the density of protuberances decreased (Supplementary Fig. 18). Instead, ordered and groove-like microstructures emerged. Correspondingly, confocal laser scanning microscopy (CLSM) measurements indicated a higher surface roughness with an arithmetical mean roughness ($S_a$) of ~21.82 μm and root mean square roughness ($S_q$) of ~25.72 μm (Fig. 2c and Supplementary Fig. 19). This roughened surface morphology would work to enhance absorption within the IR spectral band.

The Raman spectra exhibited three characteristic D (~1329 cm$^{-1}$), G (~1559 cm$^{-1}$), and 2D (~2680 cm$^{-1}$) peaks of defective graphene (Fig. 2d). The P-LIG possessed abundant defects, which could facilitate interactions with the formed HEA-NPs. Interestingly, micro-Raman spectra of HEA/LIG revealed the coexistence of two distinct forms of graphene. The Raman spectra of the protuberances indicated reduced graphene defects and the presence of metal oxides. In contrast, the spectra at the remaining positions were similar to those of the P-LIG. These finds were verified by the Raman spectra of other locations across different samples surfaces (Supplementary Fig. 20). By calculations, the crystalline sizes ($L_a$) of these two sites were ~32.5 nm and

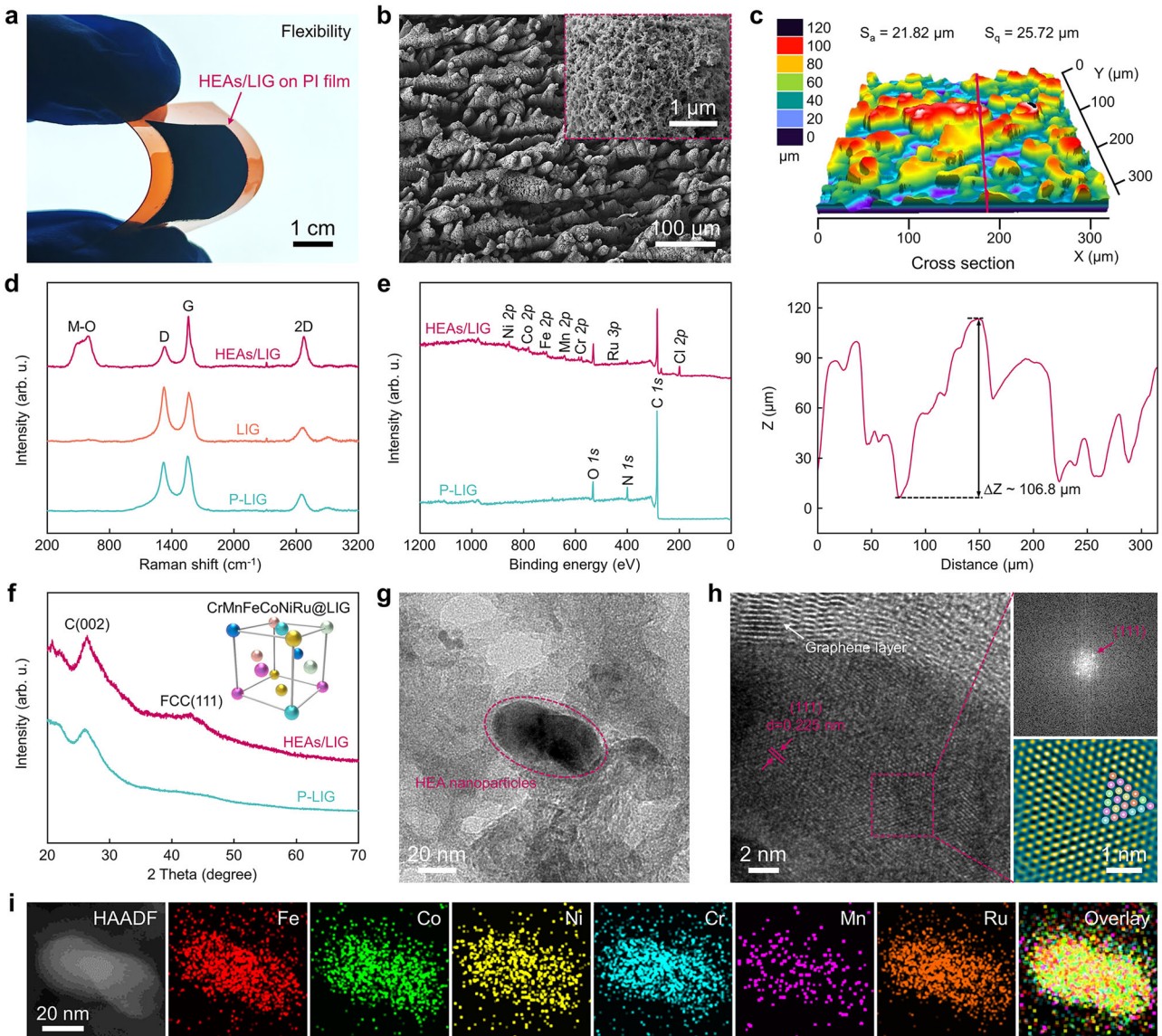

**Fig. 2 | Structural characterization of HEAs/LIG. a** Flexibility demonstration of HEAs/LIG samples. **b** SEM images of HEAs/LIG at different magnifications. **c** 3D surface morphology and cross-sectional profile of HEAs/LIG, showing the arithmetical mean roughness ($S_a$) and root mean square roughness ($S_q$). **d** Raman spectra of HEAs/LIG and P-LIG, demonstrating distinct D, G, and 2D bands as well as metal-oxide (M-O) peaks. **e** XPS survey spectra of HEAs/LIG and P-LIG. **f** XRD spectra of HEAs/LIG and P-LIG. The FeCoNiCrMnRu HEA-NPs show a face-centered cubic (FCC) structure. **g** TEM image of HEAs/LIG. **h** HRTEM, fast Fourier transform (FFT), and inverse fast Fourier transform (IFFT) images of HEA-NPs. **i** HADDF-STEM images and the corresponding EDS elemental mappings for HEA-NPs (Fe, Co, Ni, Cr, Mn, and Ru). The micrographs in (**b**) and (**g**–**i**) are representative of three independent experiments ($n$ = 3) with similar results. Source data are provided as a Source Data file.

~15.5 nm (Supplementary Note 4 and Supplementary Fig. 21). The LEAs/LIG and MEAs/LIG also exhibited analogous results but with higher defect levels in the graphene relative to the HEAs/LIG (Supplementary Fig. 22). From a perspective of surface chemistry, both energy-dispersive X-ray spectroscopy (EDS) and X-ray photoelectron spectroscopy (XPS) results further confirmed the existence of expected metal compositions (Fig. 2e and Supplementary Fig. 23). The deconvolution of O $1s$ spectra revealed that the second laser scan, while generating HEA-NPs, also resulted in the formation of few metal oxides (Supplementary Fig. 24−28). Meanwhile, a reduction in LIG oxygen defects was observed. This finding was consistent with the above Raman analysis. The X-ray diffraction (XRD) data of LIG displayed a characteristic peak (~26.2°) of the graphene C(002) plane. Differently, the HEAs/LIG had additional broad peak (~43.1°), proving the (111) plane of a face-centered cubic (FCC) structure and small grain size of HEA-NPs (Fig. 2f). The C(002) peak in the HEAs/LIG sample shifted to a

higher angle of ~26.5°, suggesting an enhanced degree of graphitization. The transmission electron microscopy (TEM) images of P-LIG and LIG further showed the typical graphene layer of ~0.34 nm (Supplementary Figs. 29, 30). In contrast, the size distribution of the FeCoNiCrMnRu HEA-NPs mainly ranged from 5 to 30 nm (Fig. 2g and Supplementary Fig. 31). There were few large-sized FeCoNiCrMnRu HEA-NPs with a ~ 200 nm size (Supplementary Fig. 32). High-resolution TEM (HRTEM) images showed that the FeCoNiCrMnRu HEA-NPs were enveloped by multilayer graphene (Fig. 2h). An interplanar spacing of 0.225 nm for the (111) plane was observed, which corresponded to the (111) plane of FeCoNiCrMnRu HEA-NPs and was in accord with the results obtained from XRD data. It is noteworthy that observed expansion in the (111) interplanar spacing of FeCoNiCrMnRu, compared to its Ru-free counterpart, serves as indirect evidence for the successful incorporation of Ru atoms into the lattice. Owing to the competition of HEA-NPs for the adsorption of oxygen in LIG, a small

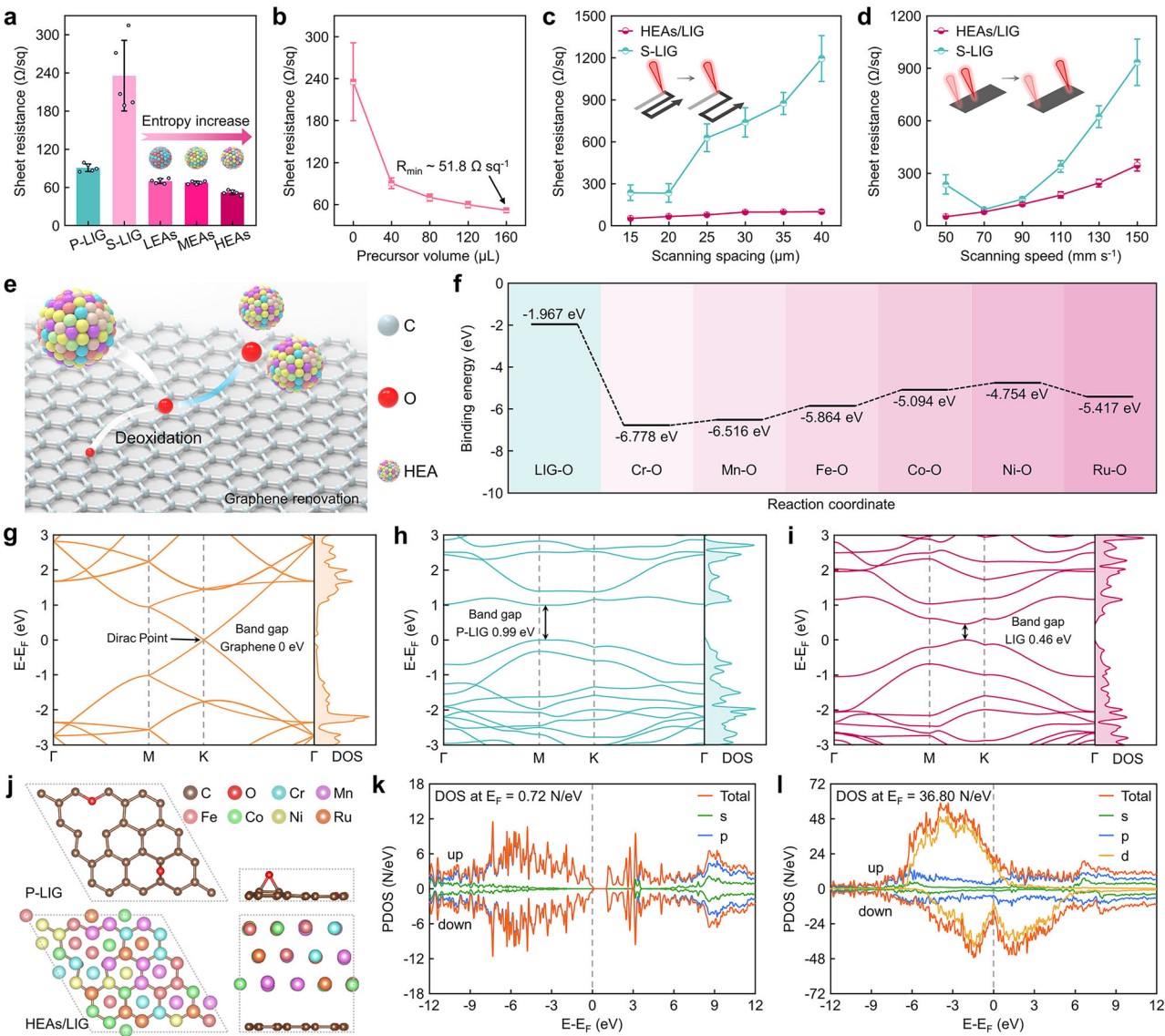

**Fig. 3 | Electrical properties of HEAs/LIG. a** Sheet resistance of P-LIG, S-LIG, LEAs, MEAs, and HEAs. Here, LEAs, MEAs and HEAs denote LEAs/LIG, MEAs/LIG and HEAs/LIG, respectively. Data are presented as mean values ± SD (*n* = 5 per group). **b** Sheet resistance of HEAs/LIG with different adding precursor volumes. Data are presented as mean values ± SD (*n* = 5 per group). Resistance variations of S-LIG and HEAs/LIG as a function of (**c**) scanning spacing and (**d**) scanning speed. Data are presented as mean values ± SD (*n* = 5 per group). **e** Schematic diagram of HEANPs for deoxidizing and renovating LIG. **f** The binding energy of deoxidation without and with various metal catalyst (Fe, Co, Ni, Cr, Mn, and Ru). Calculated band structures and density of states of (**g**) ideal graphene, (**h**) P-LIG, and (**i**) LIG (see Supplementary Fig. 39 for more modeling details). **j**, Schematic diagram of the lattice modeling for P-LIG and HEAs/LIG. Calculated DOS of (**k**) P-LIG and (**l**) HEAs/LIG. Source data are provided as a Source Data file.

amount of high-entropy oxide nanoparticles (HEO-NPs) with spinel structure were discovered (Supplementary Fig. 33). The formation of these HEO-NPs originated from the renovation of highly defective LIG by the metallic nanoparticles under the femtosecond laser-induced high-temperature and high-pressure environment. Given that the detection limit of XRD technology for crystalline phases is within the range of 1-5 wt%, the volume fraction of these surface HEOs may be well below this detection threshold. Critically, the HEO-NPs usually exhibited excellent infrared emissivity across a broad band due to efficient electron transitions and enhanced lattice vibration absorption[21]. Therefore, the transformation ultimately contributed positively to the overall infrared emissivity of the HEAs/LIG composite material. Their presence helps to enhance the overall infrared emissivity of the device, thereby improving the infrared radiation capability and Joule heating performance. The high-angle annular dark-field scanning TEM (HAADF-STEM) and element mappings indicated that the distribution

of each metallic element was highly uniform (Fig. 2i, Supplementary Figs. 34, 35). The reason for the lower Mn content compared to other elements might be attributed to the temperature during femtosecond laser treatment exceeding the boiling point of Mn, resulting in partial evaporation. Furthermore, the particle size of the FeCoNiCrMnRu HEA-NPs was found to decrease with increasing laser scanning speed (Supplementary Fig. 36). This trend can be attributed to the reduced interaction time and effective growth time for the nanoparticles at higher scanning speeds, which limits the coalescence.

## Electrical properties and electronic structure

We explored the influence of precursors and laser parameter on the sheet resistance. The direct secondary femtosecond laser scanning fabricated laser-induced graphene (S-LIG) exhibited significantly increased sheet resistance Fig. 3a). In contrast, samples added the metal precursor prior to the secondary laser scanning consistently

demonstrated sheet resistance values lower than both P-LIG and S-LIG. Among all alloy nanoparticles/LIG, the HEAs/LIG exhibited the lowest sheet resistance (-51.8 Ω sq$^{-1}$). The senary FeCoNiCrMnRu sample also showed a slightly lower sheet resistance compared to both quinary FeCoNiCrRu (-60.2 Ω sq$^{-1}$) and FeCoNiCrMn (-59.9 Ω sq$^{-1}$) (Supplementary Figs. 37, 38). Furthermore, as the precursor volume increased, the sheet resistance of HEAs/LIG decreased rapidly (Fig. 3b). Besides the precursor compositions, laser parameters also play a crucial role. Experimental results demonstrated that across all laser parameter sets, samples added with HEA precursor solution showed lower sheet resistance than the non-additive samples. Moreover, reducing both the scanning spacing and speed led to decreased sheet resistance (Fig. 3c, 3d). Narrower spacing or slower speed would increase the overlap of scanning paths, resulting in enhanced thermal accumulation and higher thermal cycling frequency. Therefore, Therefore, in order to avoid damage to the sample, we selected the parameters of 15 μm and 50 mm s$^{-1}$ for obtaining the HEAs/LIG with the lowest sheet resistance.

Then, we studied the electronic structure of HEAs/LIG using DFT calculations (see more detailed parameter settings in Supplementary Note 5). DFT calculations were conducted to evaluate the effect the interaction between HEA-NPs and LIG, as well as their impacts on conductivity. The high temperature not only produced HEA-NPs but also drove the removal of heteroatoms from LIG. In the presence of metal nanoparticles acting as catalysts, competitive binding to oxygen accelerates the shift of oxygen heteroatoms (Fig. 3e). Calculation results confirmed the binding energies between oxygen and LIG, as well as between oxygen and Fe, Co, Ni, Cr, Mn, Ru metals (Supplementary Figs. 39–44). The metal-oxygen interaction was obviously stronger than LIG-oxygen[55], indicating that free oxygen atom was preferentially adsorbed on the metals, which was consistent with Raman and XPS results (Fig. 3f). We further examined how removing oxygen heteroatoms modified the band structures of LIG (Supplementary Fig. 45). In contrast to ideal graphene with zero bandgap, the intrinsic defects made the LIG present natural semiconducting behavior (Fig. 3g). The bandgap of LIG (~ 0.46 eV) was narrower than P-LIG (~ 0.99 eV) (Fig. 3h, i), which significantly reduced the energy required for electron transitions from the valence band to the conduction band. Under excitation by an external electric field, more electrons could be excited to the conduction band, thereby effectively enhancing the conductivity. Both P-LIG and LIG were direct bandgap semiconductors, while the LIG possessed notably conduction and valence bands dispersion, leading to a smaller effective mass and enhanced conductivity. While the DFT calculation results indicate that this transformation consumes some HEAs by converting them into HEOs, resulting in a decrease in conductivity. But it also facilitates the renovation of LIG, which concurrently enhanced its overall electrical conductivity. Considering both oxygen-heteroatom removal and the loading of HEA-NPs, we calculated the density of states (DOS) of P-LIG and HEAs/LIG (Fig. 3j and Supplementary Fig. 46). Compared with P-LIG, the HEAs/LIG heterostructure showed increased electronic states near the Fermi level in the spin-up and spin-down states (Figs. 3k, l). The significant change at the Fermi level was mainly attributed to the d orbital contributions of various metal (Supplementary Fig. 47)[56]. DFT results clearly clarified the basic reason for the observed decrease in sheet resistance.

## Electrothermal conversion and infrared emissivity

The low sheet resistance of HEAs/LIG endowed it with good electrothermal conversion performance (Supplementary Note 6). Both P-LIG and HEAs/LIG exhibited linear I−V characteristics consistent, with fitted resistances of -39.7 Ω and -25.1 Ω (Fig. 4a). At 8 V voltage, the HEAs/LIG could reach -205.2 °C, whereas P-LIG only realized -146.8 °C (Fig. 4b, 4c). Infrared camera images directly showed the equilibrium temperatures at different voltages and the uniform temperature fields of P-LIG and HEAs/LIG surfaces (Supplementary Fig. 48). Moreover, a linear correlation of the equilibrium temperature with the square of voltage for both P-LIG and HEAs/LIG has been found (Supplementary Fig. 49). At a given voltage, the HEAs/LIG responded markedly faster than P-LIG (Fig. 4d). As for the high entropy samples, the FeCoNiCrMnRu showed a performance improvement in equilibrium temperatures at the 8 V voltage compared to both FeCoNiCrRu and FeCoNiCrMn. Specifically, the equilibrium temperatures of the FeCoNiCrMnRu, FeCoNiCrRu, and FeCoNiCrMn were -204.1 °C, -183.4 °C, and -185.3 °C, respectively (Supplementary Figs. 50, 51). The temperature variation of HEAs/LIG was recorded in-time at an interval of 2 V, indicating an excellent temperature response to voltage change (Fig. 4e). Under a constant voltage applied for 20 min, the surface temperature of HEAs/LIG remained nearly unchanged, demonstrating reliable stability and reliability for electric heating (Fig. 4f). Overall, the electrothermal performance of HEAs/LIG composite was highly competitive with other recent reports (Supplementary Fig. 52). We further established the relationship between equilibrium temperature and input power density, and summarized the heater energy efficiency of other advanced electrothermal materials. This comparison was intended to provide a performance standard for our prepared HEAs/LIG within the landscape of reported advanced film and flexible electrothermal devices. It was acknowledged that some of the compared devices were developed for multifunctional applications. Heater energy efficiency can be interpreted as a comparative performance metric, representing the electrical power density required to achieve a specified surface temperature under identical input conditions. As for the HEAs/LIG, the fitting calculation showed a heater energy efficiency of up to -285.4 °C cm$^2$ W$^{-1}$, indicating superior performance that surpassed most electrothermal materials (Fig. 4g and Supplementary Table 7)[57–61]. The high heater efficiency of HEAs/LIG represents an objective outcome derived after accounting for both infrared radiation and convective heat transfer. The higher heater efficiency of HEAs/LIG may also be plausibly attributed to the disruption of air convection by their surface structural morphology. The bending stability of HEAs/LIG heater was examined. We performed 200 cycles bending tests and monitored the evolution of its electrothermal performance metrics (Supplementary Fig. 53). The electrothermal response remained unchanged after testing. This exceptional durability under mechanical deformation originated from the synergistic combination of the flexible LIG substrate and the anchored HEA-NPs.

The IR emissivity quantifies the strength of the thermal radiation capacities. To investigate the radiation properties, we measured the IR emission spectra (2.5−20 μm) and emissivity of HEAs/LIG and P-LIG, with low-emissivity aluminum (Al) as a reference (Supplementary Note 7). After loading HEA-NPs, the HEAs/LIG still displayed high average emissivity across the full IR band (Fig. 4h), with the average emissivity calculated as 0.980, whereas the Al exhibited only 0.058 (Supplementary Figs. 54, 55). The infrared emissivity was consistently high across all the FeCoNiCrMnRu, FeCoNiCrRu, and FeCoNiCrMn samples, further indicating that this property is predominantly governed by the LIG carrier rather than the specific HEA composition. (Supplementary Fig. 56). Although the spectral distribution of thermal radiation varies with temperature (Wien displacement law), the broadband high emissivity across the entire infrared spectrum of HEAs/LIG samples ensures excellent radiative performance at different applied voltages. The exceptional radiative performance originated from the rougher laser-induced hierarchical microstructures with feature sizes exceeding IR wavelength. These microstructures can promote multiple reflections and absorptions of light, which greatly boosted the light-substrates interaction and achieve high emissivity (Fig. 4i). COMSOL optical simulation was adopted to investigate the electric field distribution of P-LIG, HEAs/LIG, and Al at different wavelengths (Supplementary Note 8). The simulation results suggested strong IR absorption of P-LIG and HEAs/LIG, which was consistent with our experimental measurements (Fig. 4j, Supplementary

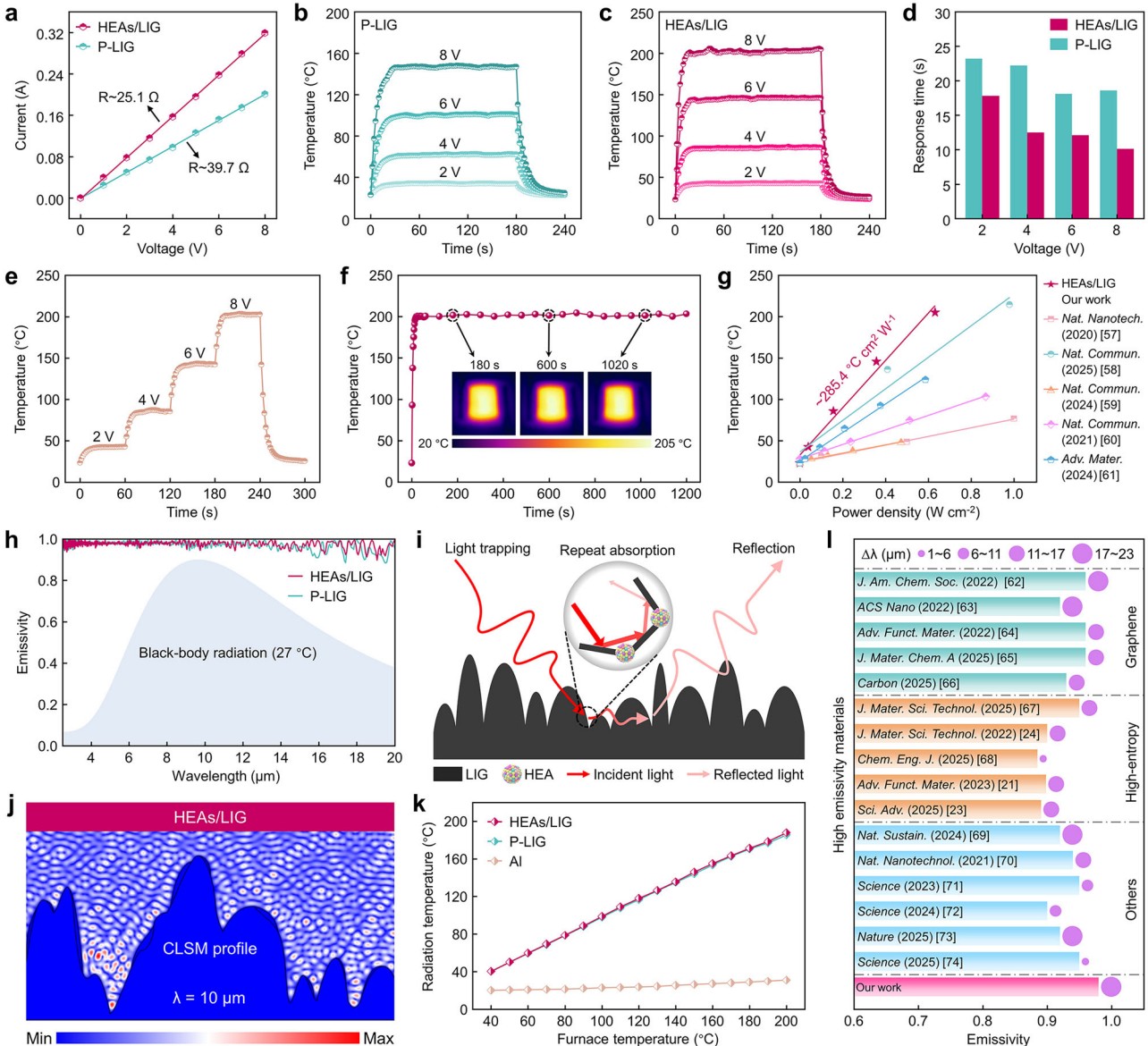

**Fig. 4 | Electrothermal conversion and infrared emissivity properties of HEAs/LIG. a** I – V linear curves of P-LIG and HEAs/LIG. Temperature-time curves of (**b**) P-LIG and (**c**) HEAs/LIG at the voltages of 2, 4, 6, and 8 V. **d** Response time of P-LIG and HEAs/LIG at different voltages. **e** Temperature curve of HEAs/LIG heater as a function of time during a stepwise voltage rising from 2 to 8 V. **f** Surface temperature-time curve of HEAs/LIG during long-term test with applying a voltage of 8 V. The size of the tested sample surface was 20 mm × 20 mm in infrared camera images. **g** Comparison of heater energy efficiency between the HEAs/LIG and various heaters (see Supplementary Table 7 for more details). **h** Infrared emissivity of P-LIG and HEAs/LIG in the wavelength range of 2.5 to 20 μm. **i** Schematic illustration for the excellent infrared absorption/emission properties of HEAs/LIG. **j** Electric field distribution of HEAs/LIG at the wavelength of 10 μm. Structural modeling was derived from confocal laser scanning microscopy (CLSM) data. **k** Radiation temperature variation trend of HEAs/LIG, P-LIG, and Al at different test temperatures. **l** Comparison of the average infrared emissivity and wavelength range of HEAs/LIG with other recently reported high emissivity materials (see Supplementary Table 8 for more details). Source data are provided as a Source Data file.

Figs. 57, 58). On a hot stage, both P-LIG and HEAs/LIG showed radiative temperatures closest to the set temperature, while the Al almost only slightly increased (Fig. 4k and Supplementary Fig. 59). Conventional high-entropy materials were usually limited in restricted band and emissivity[21]. Our HEAs/LIG composites overcame these constraints and outperformed most high emissivity materials, including graphene[62–66], high-entropy[21,23,24,67,68], and other materials[69–74] (Fig. 4l and Supplementary Table 8)[23,52,53]. Furthermore, the HEAs/LIG samples also showed excellent durability and stability during the air exposure, high humidity, and repeated heating/cooling test. The sheet resistance and surface emissivity of both tested samples almost remained unchanged after stability test (Supplementary Fig. 60). This exceptional oxidation resistance was attributed to the unique core-shell structure formed

during our femtosecond laser processing (Supplementary Fig. 61). The formed graphene shell encapsulating the HEA-NPs provides an excellent barrier against oxidation and moisture (Supplementary Fig. 62). The porous and robust LIG substrate firmly anchors the HEA-NPs, preventing their migration and agglomeration under repeated thermal treatment. Contact angle measurements revealed the hydrophobicity of P-LIG and HEAs/LIG with water contact angles of 152.8° and 145.3°, which further expanded their multifunctionality (Supplementary Figs. 63, 64).

## High-performance Joule heating

Finally, experiments validated the exceptional radiative heating ability of HEAs/LIG (Fig. 5a, Supplementary Figs. 65–68). Due to the high

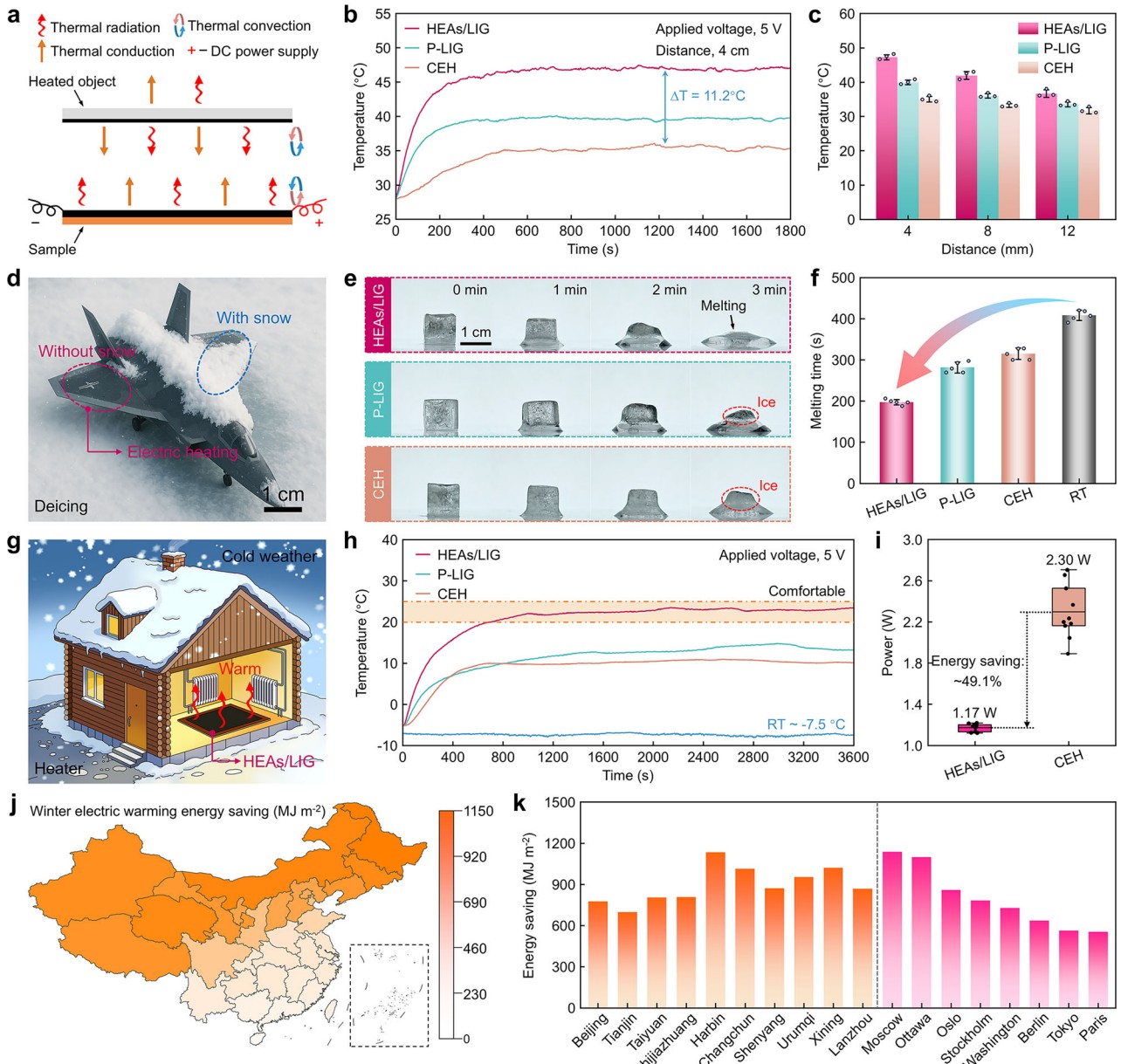

**Fig. 5 | Joule heating evaluation and potential applications of HEAs/LIG.**
**a** Diagram describing the thermal transfer in the radiative thermal transfer experiment. **b** Temperature-time curves of the upper surface of heated object when radiated by HEAs/LIG, P-LIG, and commercial electric heater (CEH), respectively. **c** Saturation temperature of heated object with different distances between heated object and heater. Data are presented as mean values ± SD (n = 3 per group). **d** Optical diagram for HEAs/LIG using for deicing on the aircraft in winter. **e** Optical photos of ice during radiative heating deicing test. **f** Time for complete melting of ice with different heaters heating. RT: spontaneous ice melting at room temperature without heating. **g** Schematic diagram of HEAs/LIG for electric warming in cold environments. Data are presented as mean values ± SD (n = 5 per group). **h** Temperature changes of the heated object inside the house when radiated by

HEAs/LIG, P-LIG, and CEH, respectively. **i** Box plots of the power required for HEAs/LIG and CEH heaters to achieve comfortable conditions. The center lines represent the mean values, box bounds indicate the 25th and 75th percentiles, and whiskers extend to the minimum and maximum values. Individual data points are shown as dots (n = 10 per group). **j** Winter warming energy saving of HEAs/LIG compared with CEH. Basemap in Fig. 5j derived from Runfola, D. et al., geoBoundaries: A global database of political administrative boundaries (https://doi.org/10.1371/journal.pone.0231866)[75], which were provided under the CC BY 4.0 license (https://creativecommons.org/licenses/by/4.0/). No changes were made. **k** Winter warming energy saving in different cities of China and some cities in the world. Source data are provided as a Source Data file.

heater energy efficiency and infrared emissivity, the HEAs/LIG could achieve a higher surface temperature under the same electrical power input. This elevated temperature, coupled with the high emissivity, enables efficient radiative heating of objects. The real-time temperature curves of heated object indicated that HEA/LIG could maintain a steady-state temperature ~11.2 °C higher than CEH with the distance of 4 mm and 5 V voltage (Fig. 5b). At different distances, the HEAs/LIG expectedly achieved the highest stable temperature and fastest

heating speed, demonstrating superior heating performance (Fig. 5c, Supplementary Figs. 69, 70). This was due to the broadband high IR emissivity and superior electrothermal conversion performance. It was worth noting that the HEAs/LIG heater showed excellent stability during the rapid on/off cycling test (Supplementary Fig. 71). The reliable cycling stability comes from the robust interface between the HEA-NPs and the LIG substrate, as well as the intrinsic high-temperature stability and anti-oxidation properties of the graphene

shell-encapsulated HEA-NPs composition. Similarly, HEAs/LIG could be applied to de-icing, snow removal, and warming in cold environments (Fig. 5d). The HEAs/LIG heater realized complete ice melting within 3 min, demonstrating excellent melting efficiency (Fig. 5e, 5f, Supplementary Fig. 72, and Movie 1). Here, RT indicates that the ice melting occurred naturally at room temperature with no heater used. In cold environments, HEAs/LIG heater also enabled energy-efficient personal warming (Fig. 5g). At 5 V voltage, the HEAs/LIG heater could lift the upper surface temperature of the heated object from ~−7.5 °C to a human-comfortable level of ~24.0 °C, while CEH failed under the same conditions (Fig. 5h and Supplementary Fig. 73). Crucially, the HEAs/LIG required only ~1.17 W to maintain thermal comfort, whereas the CEH demanded ~2.3 W, yielding a notable ~49.1% energy saving (Fig. 5i). Under the tested specific geometric and enclosed conditions (housing model), HEAs/LIG heaters achieved a measurable reduction in the electrical energy consumption required to reach the same target object temperature, thereby demonstrating an energy-saving effect. Energy saving efficiency of HEAs/LIG in winter was further evaluated[75,76]. The energy savings of provincial capitals in China was given in Fig. 5j. The detailed assumptions, parameters, and methodology of the energy saving calculation could be found in Supplementary Note 9. Using the computational models, we mainly calculated the winter energy savings potential across China. Compared to using CEH, HEAs/LIG heater delivers markedly higher energy saving performance. For example, Harbin can save roughly 1134.99 MJ m$^{-2}$. The projected energy savings of some Chinese and global cities further prove the potential of HEAs/LIG heater (Fig. 5k). The experimental and complementary simulations prove that the HEAs/LIG heater offers an exceptional pathway toward sustainable thermal management, positioning them as strong candidates for energy-efficient heating technologies.

## Discussion

We achieved the successful fabrication of FeCoNiCrMnRu HEA-NPs on LIG by using femtosecond laser solid-phase synthesis. The rapid heating and cooling processes induced by femtosecond laser on a highly absorptive LIG substrate facilitated the formation of HEA-NPs. Comprehensive structural and DFT calculations analyses revealed the mechanism for enhancing conductivity was the loading of HEA-NPs and their deoxygenation renovation effect. Remarkably, the resulting HEAs/LIG functions as a superior Joule heater, delivering a heater energy efficiency of ~285.4 °C cm$^2$ W$^{-1}$ and average infrared emissivity of ~0.98 across a wide wavelength range of 2.5−20 μm. Finally, we verified that the HEA/LIG as an electric heater in winter could realize energy saving of ~49.1% relative to conventional CEH. These results underscore the potential of HEAs/LIG as high-performance Joule heating materials, enlarging the application scope and function of HEAs.

## Methods

### Materials and reagents

Polyimide (PI) film with a thickness of 200 μm was supplied by Zhongshan Chenxi Electronics Co., Ltd. (Guangdong, China). Titanium (Ti) substrates with a thickness of 1 mm and aluminum (Al) sheets were bought from Anhui Zhengying Technology Co., Ltd (Anhui, China). The plane model (Fig. 5d) and house model (Supplementary Fig. 73) were purchased from Taobao e-commerce platform. Absolute alcohol, FeCl$_3$·6H$_2$O (>99%), CoCl$_2$·6H$_2$O (>99.9%), NiCl$_2$·6H$_2$O (>99.9%), CrCl$_3$·6H$_2$O (>98%), MnCl$_2$·4H$_2$O (>99%), and RuCl$_3$·xH$_2$O (>98%) were purchased from Shanghai Macklin Biochemical Co., Ltd (Shanghai, China). Contact-angle testing on all materials was performed using deionized water. Commercial electric heaters (CEH) with the size of 20 mm × 20 mm × 2 mm were purchased online. The CEH was covered with Al. Therefore, Al sheets were chosen as a reference to test the infrared (IR) emissivity. The laser-treated Ti substrate was used as the heated object in the radiative thermal transfer experiment (Figs. 5b

and 5h). All chemicals were used as received without further purification.

### Preparation of P-LIG

Pristine LIG (P-LIG) was prepared using femtosecond laser direct writing technology. The laser beam (pulse width: 350 fs; central wavelength: 1035 nm; repetition rate: 400 kHz) originated from a commercial femtosecond fiber laser (HR-Femto-IR-50-40B, Huaray, China) and was used to perform ablation operations. During the processing, the laser beam was directed onto the PI film surface via a two-mirror galvanometric scanning system (basiCube 10, Scanlab, Germany) equipped with an f-θ lens (focal length 125 mm), thereby achieving laser focusing and scanning along the x-y axis directions. For P-LIG preparation, the laser scanning spacing was set to 15 μm, power to 2.6 W, and scanning speed to 100 mm s$^{-1}$. The size of the as-prepared sample area was 20 mm × 20 mm.

### Precursor loading on P-LIG

In a typical preparation process, various metal chlorides were mixed in absolute alcohol at 0.1 M for each metallic element. The mixed solution was directly dropped onto the P-LIG surface with different volume. Then, the P-LIG with precursor was dried at room temperature.

### Preparation of LEAs/LIG, MEAs/LIG, HEAs/LIG, and S-LIG

Then, the P-LIG with precursor were was irradiated by the same laser as mentioned above. The laser synthesis was conducted with a variation of scanning spacing (15, 20, 25, 30, 35, and 40 μm) and scanning speed (50, 70, 90, 110, 130, and 150 mm s$^{-1}$), and fixed laser power of 2.0 W. Finally, the FeCoNiCrMnRu HEA-NPs loaded on the LIG (HEAs/LIG) were obtained. Similarly, the samples prepared using low-entropy (FeCo) and medium-entropy (FeCoNiCr) mixed precursor solutions were named LEAs/LIG and MEAs/LIG, respectively. Moreover, the high-entropy samples (FeCoNiCrRu and FeCoNiCrMn) were prepared to verify the influence of Ru and configurational entropy on the Joule heating performance. Samples subjected to a second direct writing scan using the femtosecond laser without dispensing the precursor solution are named as secondary processed LIG (S-LIG). Furthermore, the LIG referred to the laser-induced graphene section in the HEAs/LIG composite materials. Unless otherwise specified, the scanning spacing and scanning speed adopted in this work were 15 μm and 50 mm s$^{-1}$, respectively.

### Characterization

The morphology and microstructure of the samples were examined by a field emission scanning electron microscope (FESEM, MIRA3 LMU, TESCAN, Czech Republic) equipped with an energy-dispersive X-ray spectroscopy (EDS, TESCAN, Czech Republic). The 3D surface morphology, cross-sectional profile and surface roughness were characterized by a confocal laser scanning microscope (CLSM, Axio LSM700, Zeiss, Germany). A field-emission transmission electron microscope (FETEM, Tecnai G2 F20 S-TWIN, FEI, America) with EDS was utilized to study the morphology and elemental distribution of the as-prepared samples with an acceleration voltage of 200 kV. The elemental states on the sample surface were analyzed via an X-ray photoelectron spectroscopy (XPS, Thermo Scientific, K-Alpha, America) with a monochromatic Al Kα X-ray radiation. The crystal structure of samples was characterized by an X-ray diffractometer (XRD, XRD-6100, Shimadzu, Japan) with a scanning speed of 1 °/min using the Cu Kα radiation (λ = 1.54 Å). The Raman spectroscopy was collected with a Invia Qontor Raman Spectrometer (Renishaw, UK) and the excitation wavelength was 532 nm. The reflectance and transmittance spectra were measured by an ultraviolet−visible spectrophotometer (UV-2600, Shimadzu, Japan) equipped with a diffuse integrating sphere. The IR spectra of samples in the range of 2.5−25 μm was measured by a Bruker Vertex 70 Fourier transform infrared (FTIR) spectrometer with

an integrating sphere (A562). The emissivity of mid-infrared band (2.5 – 20 μm) was measured by a Fourier transform infrared spectrometer with a gold integrating sphere (PIKE). The microstructure of samples was observed by a metallurgical microscope (CX40M-T/RT, Ningbo Sunny Instruments Co., Ltd., China). Contact angle and dynamic wetting behavior were recorded using a contact angle goniometer (SDC-200S, Sindin) at ambient temperature. A direct current (DC) power source (SPPS1203D, Shenzhen Kuaiqu Electronics Co., Ltd., China) was applied to performing the electric heating experiments. The surface temperature distribution was investigated using an infrared thermal camera (UTi320E, UNI-T). The temperature data was acquired by a multi-channel temperature collecting system (UT3208, UNIT, China). The temperature change during femtosecond laser irradiation and electric field intensity of sample surfaces were studied using the finite element COMSOL Multiphysics software. Spin-polarized density functional theory (DFT) calculations were performed using the CASTEP module within the Materials Studio platform. All error bars in SD (standard deviation) are obtained by statistically repeating independent identical experimental results at least three times, and data are presented as mean values ± SD.

### Thermal radiation measurements

Radiative heating experiments were performed using different samples (including HEAs/LIG, P-LIG, and CEH), with laser-treated Ti as the object to be heated (Supplementary Fig. 66). A DC power supply was used to apply direct current to the electric heater, and the HEAs/LIG side faced the surface of the laser-treated Ti, while the temperature on the back side of the laser-treated Ti was recorded using a multi-channel temperature collecting system (Fig. 5a and Supplementary Fig. 65). The temperature-time curves at different distances between heated object and heater was also measured to obtain heating rates by curve fitting. For Fig. 5e, an ice cube was used as the heated object for radiative melting tests. For Fig. 5g, radiative space-heating tests were conducted in a closed house (house model shown in Supplementary Fig. 73) under an ambient temperature of −7.5 °C following the same procedure as Fig. 5a with the back-side temperature of the laser-treated Ti recorded. For Fig. 5i, the steady-state electrical power required for the HEAs/LIG and CEH heaters to heat the object to a comfortable temperature was measured.

## Data availability

All data generated in this study are provided in the Source Data file or are available from the corresponding author upon request. Source data are provided with this paper.

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

## Acknowledgements

This research was supported by the National Natural Science Foundation of China (Grant Nos. 92580114, 52475499, and 52222513 to K.Y., and 525B2076 to L.W.), National Key R&D Program of China (Grant No. 2023YFB4604200 to K.Y.), Natural Science Foundation of Hunan Province (Grant No. 2025JJ30016 to K.Y.), the State Key Laboratory of High-performance Precision Manufacturing (Grant No. HPMKF202505 to K.Y.), Fundamental Research Funds for the Central Universities of Central South University (Grant Nos. 2025ZZTS0096 and 2025ZZTS0680 to

K.Y.). The dataset used in Fig. 5i derived from geoBoundaries (https://www.geoboundaries.org). This work was supported in part by the High Performance Computing Center of Central South University.

## Author contributions

L.W. conceived the idea, designed the experiments, performed experimental studies, organized the data, and wrote the manuscript. K.Y. conceived the idea, designed the experiments, overseen the data, provided scientific guidance, provided experimental resources, and revised the manuscript. J.X., X.S., and J.P. contributed to the characterization and discussion of data. K.Y., J.H., and J.A.D. supervised all aspects of the research. J.H. and J.A.D. provided some experimental resources. All authors discussed the results, made comments, and approved the manuscript.

## Competing interests

The authors declare no competing interests
