## [Transparent Peer Review file · Nature Communications]

Femtosecond laser synthesis of multiscale high-entropy alloys/graphene composites for high-performance Joule heating

Corresponding Author: Professor Kai Yin

Version 0:

Reviewer comments:

Reviewer #1

(Remarks to the Author)

The manuscript reports the femtosecond-laser, in-situ synthesis of FeCoNiCrMnRu high-entropy alloy (HEA) nanoparticles on laser-induced graphene (LIG), forming a multiscale HEA/LIG hybrid for electro-thermal heating and broadband infrared emission. However, several critical elements of the evidence chain are incomplete. I therefore recommend major revision.

1. Air exposure, high humidity, and repeated heating/cooling can lead to oxidation, migration, or agglomeration of metal nanoparticles, affecting electrical and radiative properties. Stability under these conditions should be substantiated.
2. The presence of HEO observed by TEM should be quantified. The manuscript should clarify their fraction and discuss their influence on emissivity and conductivity, and thus on the overall device performance.
3. Where blackbody radiation curves are shown (e.g., Fig. 4h), the corresponding blackbody temperature should be explicitly annotated.
4. Thermal/optical modeling requires clear disclosure of material constants (e.g., ρ , c_p , k , n , κ) and boundary conditions, along with a brief uncertainty/sensitivity discussion.
5. Given the local nature of femtosecond direct writing, the manuscript should address large-area uniformity, batch-to-batch reproducibility, and practical scalability/throughput considerations.
6. Bending/adhesion/abrasion durability data are important for flexible device claims and should be presented with clear before/after performance metrics.
7. If energy-efficiency advantages over commercial heaters are asserted, the underlying assumptions, parameters, and calculation approach should be transparent.
8. The use of Ru raises questions about cost and supply. A brief discussion of material criticality and potential substitutions would improve the practicality narrative.
9. A direct comparison between FeCoNiCrMn (high-entropy) and FeCoNiCrMnRu is needed to reveal whether performance gains arise from Ru itself or from marginally higher configurational entropy.
10. For Joule-heating use cases, rapid on/off cycling stability should be demonstrated, with tracking of sheet resistance, heating metrics, and emissivity over extended cycles.

Reviewer #2

(Remarks to the Author)

This manuscript presents the synthesis of high-entropy alloy nanoparticles (HEA-NPs) on laser-induced graphene (LIG) using femtosecond (fs) laser irradiation, aiming to demonstrate enhanced Joule heating performance. However, This reviewer finds several fundamental concerns that undermine the validity and significance of the reported results. The justification for using a femtosecond laser is insufficient. While the authors claim a thermal mechanism for HEA nanoparticle formation, femtosecond lasers are typically associated with non-thermal, "cold" ablation processes. The manuscript does not demonstrate clear advantages of fs lasers over more accessible alternatives like nanosecond or CW lasers, raising concerns about the method's practicality and relevance.

The thermal simulation based on the classical heat diffusion equation is also problematic. It fails to capture ultrafast non-equilibrium dynamics occurring in the first few picoseconds after fs-laser exposure, leading to questionable predictions such as a ~3000 K peak temperature and a cooling rate of $\sim 5.6 \times 10^{11}$ K/s. Figures 1b–d rely heavily on this model, without addressing material phase stability or the validity of such extreme thermal conditions. Key physical effects such as laser ablation, evident in Fig. S13, and heat accumulation from pulse overlap are neglected in the simulation. These omissions limit the credibility of the thermal analysis and the claimed mechanism of HEA nanoparticle formation.

Lastly, the metric of “electrothermal efficiency” appears equivalent to area-normalized thermal resistance, which may overstate performance for thinner films. The comparison in Fig. 4g seems not fair, as many cited devices were designed for multifunctionality, not optimized heating.

Given the above issues, this reviewer finds that the manuscript contains critical scientific flaws. Therefore, unfortunately, this manuscript is not recommended for publication.

Reviewer #3

(Remarks to the Author)

In this manuscript, they achieved the successful fabrication of FeCoNiCrMnRu HEA-NPs on LIG by using femtosecond laser ultrafast solid-phase synthesis. It is a novel research. Some parts should be revised. It can be accepted after revisions.

1. Pristine LIG (P-LIG) was prepared using femtosecond laser direct writing technology. How much of the yield of the nanoparticles should be mentioned? How many grams did you prepared?

2. How to control the particle size to make it uniform by so fast laser processing. A TEM image of the nano HEA particles should be added.

3. How can you find it for high-performance Joule heating applications? occasionally?

4. Why did you choose FeCoNiCrMnRu? Ru is very expensive? Why must you add Ru? How about the property without Ru?

5. High-performance Joule heating applications of the eCoNiCrMnRu HEA-NPs should consider the oxidation problems of the particles. It will be oxidized quickly in air,

Version 1:

Reviewer comments:

Reviewer #1

(Remarks to the Author)

The authors have addressed all concerns satisfactorily. Recommend acceptance for publication.

Reviewer #2

(Remarks to the Author)

The authors' revised discussion on femtosecond laser processing adopts a more conservative tone and demonstrates an acceptable academic depth compared with similar recent reports.

The authors' response helps understand the rationale behind employing $E_T = 1/(h_c + h_r)$ as a normalized and comparative parameter for electrothermal performance. However, the terminology “energy-use efficiency” or “electrothermal conversion efficiency” used in the manuscript is then conceptually ambiguous. As Joule heating intrinsically converts electrical energy into heat with nearly 100% efficiency, E_T cannot be regarded as an energy-conversion efficiency in the thermodynamic sense.

Regarding this point, the “energy saving” demonstration in Fig. 5(g–i) requires further clarification. Given that Joule heating converts electricity into heat at nearly 100% efficiency regardless of heater type, the generated heat should be transferred to the surroundings through radiation and convection. As acknowledged by the authors, E_T corresponds to the inverse of the effective heat-transfer coefficient. Thus, the local $\Delta T/P$ ratio merely indicates that the film has reduced heat loss to its immediate surroundings. In this study, however, the stated purpose of the heater is to warm the environment by transferring heat outward. Consequently, a heater with a high E_T value acts as a good thermal insulator, not necessarily as an effective space heater.

It is therefore essential to clarify whether the temperature reported in Fig. 5h represents the surface temperature of the heater or the air temperature inside the house model. Furthermore, while the study claims high emissivity for the HEA heater, high emissivity would increase h_r and consequently lower E_T . This consideration appears inconsistent with the high “electrothermal conversion efficiency” for the HEA heater.

If the authors are unable to provide a convincing rebuttal to the above concerns, it indicates that the original novelty and conceptual foundation of the work are not sufficiently persuasive.

Reviewer #3

(Remarks to the Author)

I am satisfied with the revision. It can be accepted now.

Version 2:

Reviewer comments:

Reviewer #1

(Remarks to the Author)

After carefully reviewing the latest comments from Reviewer #2 together with the authors' rebuttal, I am inclined to support acceptance of the manuscript, provided that the conclusions are interpreted within clearly defined boundaries. In my assessment, the remaining disagreement is primarily related to the scope and wording of the interpretation rather than to the validity of the experimental data or the scientific soundness of the work. The study convincingly demonstrates a novel femtosecond-laser enabled route for fabricating high-entropy alloy nanoparticles on LIG and establishes their ultrahigh emissivity and effective radiative heat delivery to a nearby target object. When "heater energy efficiency (HEE)" is understood as a comparative performance metric, namely, the electrical power required to achieve a specified heating outcome under identical conditions, there is no fundamental physical inconsistency between ultrahigh emissivity and the reported heating performance. Indeed, the "heater energy efficiency" metric is a widely recognized and commonly adopted parameter for comparing the energy utilization efficiency of various electro-thermal conversion devices, such as *Adv. Funct. Mater.* 2023, 33, 2213357, *Adv. Eng. Mater.* 2022, 24, 2200368, and *Small*, 2022, 18, 2202906. The method for calculating the "heater energy efficiency" adopted in the manuscript is also consistent with those reported in other literature. Regarding the "energy saving" and heat-transfer interpretation, I agree that the manuscript should not be read as demonstrating improved general space-heating efficiency. However, the experimental data robustly support a more focused and well-defined conclusion: the proposed radiative heating approach achieves measurable reductions in electrical power consumption required to reach the same target-object temperature, within the specific geometric and enclosed conditions tested. The authors have explicitly confined the "energy saving" claim to the aforementioned conditions. The energy-saving performance of the HEAs/LIG composites has been experimentally verified, and its significant energy-saving potential has further been validated through simulations. Additionally, the authors reasonably suggest that the LIG morphology contributes to suppressing convection, thereby maintaining excellent "heater energy efficiency" even under conditions of high infrared emissivity, which should be regarded as a plausible contributing factor. Considering the above points collectively, I believe the manuscript is scientifically robust. With such restrained interpretation and minor textual refinement, the work makes a meaningful contribution to radiative heating materials and can be suitable for publication.

Reviewer #2

(Remarks to the Author)

This reviewer appreciates the multidisciplinary nature of the study and its contributions to femtosecond-laser synthesis of high-entropy alloy nanoparticles and broadband emissivity control. However, the claimed coexistence of ultrahigh emissivity and high heater energy efficiency (HEE) remains supported only qualitatively, and the demonstrated "energy saving" should be interpreted strictly in the context of radiative heating of a target object rather than as an improvement in general space-heating efficiency.

In particular, the assertion that the LIG surface morphology suppresses convection sufficiently to compensate for increased radiative losses is not convincingly validated. This is a nontrivial heat-transfer argument, and the cited literature is not directly related to this problem. Moreover, within a small "house model" enclosure, buoyancy-driven natural convection is difficult to suppress in practice. As such, the concept of an LIG-based space heater remains scientifically fragile and open to further debate.

The Response to the Reviewer's Comments

Manuscript Number: NCOMMS-25-66444

Title: Ultrafast achieving multiscale high-entropy alloys/graphene for high-performance Joule heating

Authors: Lingxiao Wang, Kai Yin, Jianqiang Xiao, Xinghao Song, Jiaqing Pei, Jun He, Ji-An Duan

We are indebted to the reviewers for their time and effort in reviewing our paper. The detailed comments from the referees are very impressive and greatly valued. We are happy to have such excellent advice for improving the paper's quality. With our best efforts and best knowledge, we have made changes by following the referees' comments. This report summarizes our understanding of the recommendations and the changes we have made in response. The reviewers' comments are reproduced following by the corresponding modifications to the manuscript.

In this report, some reference numbers, equations, and figures are marked by an R, for example, Figure R1, to distinguish them from those numbers for Figures in our original paper, for examples, Figure 1 and Figure S1. Different descriptions are also marked with different colors: Reviewer's comments are black; Our response is marked with blue and red; The part involved the revised manuscript in our response is red.

REVIEWER #1

Reviewer's Comments

The manuscript reports the femtosecond-laser, in-situ synthesis of FeCoNiCrMnRu high-entropy alloy (HEA) nanoparticles on laser-induced graphene (LIG), forming a multiscale HEA/LIG hybrid for electro-thermal heating and broadband infrared emission. However, several critical elements of the evidence chain are incomplete. I therefore recommend major revision.

Our Response

We greatly appreciate the reviewer's positive comments on our manuscript. These insightful suggestions/comments will significantly help to improve the scientific quality and presentation of this work. Below are our detailed responses to each raised concern and suggestion.

Reviewer's Comments

1. Air exposure, high humidity, and repeated heating/cooling can lead to oxidation, migration, or agglomeration of metal nanoparticles, affecting electrical and radiative properties. Stability under these conditions should be substantiated.

Our Response

We sincerely appreciate the reviewer's professional comment regarding the stability of HEAs/LIG samples under air, humidity, and thermal cycling conditions. These factors including oxidation, migration, and agglomeration are critical challenges that can degrade electrical and radiative properties. In our revised manuscript, we have taken a comprehensive stability assessment to substantiate the robustness of our FeCoNiCrMnRu HEAs/LIG. The emissivity of mid-infrared band (2.5–20 μm) was measured by a Fourier transform infrared spectrometer with a gold integrating sphere (PIKE).

Specifically, we compared the electrical and radiative properties of the pristine HEAs/LIG sample with two tested samples. A sample stored for six months under ambient conditions with the temperature 20–25 °C and relative humidity of ~70%, while the other subjected to extensive Joule heating-cooling cycles after 100 cycles. We found that both the aged and thermally cycled samples retained their electrical and radiative properties with negligible change compared to the pristine sample. The sheet resistance and surface emissivity of both tested samples almost remained unchanged after stability test (**Figure R1**). Then, we performed detailed TEM and EDS analysis on these aged samples (**Figure R2**). This exceptional oxidation resistance was attributed to the unique core-shell structure formed during our femtosecond laser processing (**Figure R3**). The formed graphene shell encapsulating the HEA-NPs provides an excellent barrier against oxidation and moisture (**Figure R4**) [*Adv. Mater.*, 2024, 36, 2402391]. The porous and robust LIG matrix firmly anchors the HEA-NPs, preventing their migration and agglomeration under repeated thermal treatment. These comprehensive stability data strongly verify the durability of the HEAs/LIG.

Figure R1. (a) Sheet resistance and (b) infrared emissivity of the HEAs/LIG in the wavelength range of 2.5 to 20 μm before and after stability test. A sample stored for eight months under ambient conditions with the temperature 20–25 °C and relative humidity of ~70%, while the other subjected to extensive Joule heating-cooling cycles after 100 cycles.

Figure R2. TEM images of HEA-NPs after stability test.

Figure R3. HRTEM images of HEA-NPs after stability test, showing the multilayer graphene encapsulating the HEA-NPs.

Figure R4. HAADF-STEM images and the corresponding EDS elemental mappings for HEA-NPs (Fe, Co, Ni, Cr, Mn, Ru, and O) after stability test.

We have added a discussion about the stability of HEA-NPs in the revised manuscript. We also included **Figure R1** as **Figure S60**, **Figure R3** as **Figure S61**, and **Figure R4** as **Figure S62** in the supplementary information, as follows:

Furthermore, the HEAs/LIG samples also showed excellent durability and stability during the air exposure, high humidity, and repeated heating/cooling test. The sheet resistance and surface

emissivity of both tested samples almost remained unchanged after stability test (Supplementary Fig. 60). This exceptional oxidation resistance was attributed to the unique core-shell structure formed during our femtosecond laser processing (Supplementary Fig. 61). The formed graphene shell encapsulating the HEA-NPs provides an excellent barrier against oxidation and moisture (Supplementary Fig. 62). The porous and robust LIG substrate firmly anchors the HEA-NPs, preventing their migration and agglomeration under repeated thermal treatment. (Page 13, Manuscript)

Reviewer's Comments

2. The presence of HEO observed by TEM should be quantified. The manuscript should clarify their fraction and discuss their influence on emissivity and conductivity, and thus on the overall device performance.

Our Response

We sincerely appreciate the reviewer's insightful observation regarding the HEOs detected in our TEM analysis. Furthermore, we agree that a detailed discussion on their fraction and influence is crucial for understanding the device performance. In our experiments, TEM and Raman data confirmed the presence of trace HEOs (**Figure R5**), while XRD showed no discernible crystalline oxide peaks (**Figure R6**). Providing an exact quantitative fraction of these oxides from TEM alone is challenging due to the statistical limitations. However, based on the extensive TEM observation, Raman, and XRD results, we can provide a qualitative estimate of the presence of HEOs.

Our TEM images detected the presence of HEOs. The Raman spectroscopy data, which was highly sensitive to surface species and chemical bonds, provided complementary evidence for the presence of metal-oxygen bonds, further supporting the TEM observations. In view of the locality of TEM and sensitivity of Raman, we estimate that the primary reason for the absence of distinct metal oxide peaks in the XRD pattern is their exceptionally low content and high degree of dispersion. Given that the detection limit of XRD technology for crystalline

phases is within the range of 1-5 wt%, the volume fraction of these surface HEOs may be well below this detection threshold.

The formation of these HEOs originated from the renovation of highly defective LIG by the metallic nanoparticles under the femtosecond laser-induced high-temperature and high-pressure environment. While the DFT calculation results indicate that this transformation consumes some HEAs by converting them into HEOs, resulting in a decrease in conductivity [ACS Nano, 2024, 18, 34492-34530]. But it also facilitates the renovation of LIG, which concurrently enhanced its overall electrical conductivity. Critically, related literature [Adv. Funct. Mater., 2023, 33, 2303197; Adv. Mater., 2025, e08636] confirms that the HEOs usually exhibited excellent infrared emissivity across a broad band due to efficient electron transitions and enhanced lattice vibration absorption. Therefore, the transformation ultimately contributes positively to the overall infrared emissivity of the HEAs/LIG composite material. Therefore, we believe that trace HEOs are generally beneficial within our HEAs/LIG. Their presence helps to enhance the overall infrared emissivity of the device, thereby improving the infrared radiation capability and Joule heating performance.

Figure R5. Raman spectra of HEAs/LIG and P-LIG.

Figure R6. XRD spectra of HEAs/LIG and P-LIG.

We have also added corresponding discussion about the influence of HEOs in the revised manuscript:

The formation of these HEO-NPs originated from the renovation of highly defective LIG by the metallic nanoparticles under the femtosecond laser-induced high-temperature and high-pressure environment. Given that the detection limit of XRD technology for crystalline phases is within the range of 1-5 wt%, the volume fraction of these surface HEOs may be well below this detection threshold. Critically, the HEO-NPs usually exhibited excellent infrared emissivity across a broad band due to efficient electron transitions and enhanced lattice vibration absorption²¹. Therefore, the transformation ultimately contributed positively to the overall infrared emissivity of the HEAs/LIG composite material. Their presence helps to enhance the overall infrared emissivity of the device, thereby improving the infrared radiation capability and Joule heating performance. (Page 8, Manuscript)

While the DFT calculation results indicate that this transformation consumes some HEAs by converting them into HEOs, resulting in a decrease in conductivity. But it also facilitates the renovation of LIG, which concurrently enhanced its overall electrical conductivity. (Page 10, Manuscript)

Reviewer's Comments

3. Where blackbody radiation curves are shown (e.g., Fig. 4h), the corresponding blackbody temperature should be explicitly annotated.

Our Response

We sincerely thank the reviewer for the insightful comments. We have annotated the corresponding blackbody temperature for **Figure 4h** in the manuscript, **Figure S45**, and **Figure S48** in the supplementary information, as follows:

Fig. 4h, Infrared emissivity of P-LIG and HEAs/LIG in the wavelength range of 2.5 to 20 μm .

Supplementary Figure 54. Infrared emissivity of Al in the wavelength range of 2.5 to 20 μm .

Supplementary Figure 58. Simulated infrared emissivity of (a) P-LIG, HEAs/LIG, and (b) Al in the wavelength range of 3 to 20 μm .

In addition, we have also added corresponding explanations in the revised manuscript:

Although the spectral distribution of thermal radiation varies with temperature (Wien displacement law), the broadband high emissivity across the entire infrared spectrum of HEAs/LIG samples ensures excellent radiative performance at different applied voltages. (Page 13, Manuscript)

Reviewer's Comments

4. Thermal/optical modeling requires clear disclosure of material constants (e.g., ρ , c_p , k , n , k) and boundary conditions, along with a brief uncertainty/sensitivity discussion.

Our Response

We appreciate the reviewer's rigorous attention to the methodological details of our simulation framework. We have now supplemented the manuscript with a detailed description of the parameters, boundary conditions, and an uncertainty analysis related to our thermal/optical modeling. The revisions can be found in the revised supplementary information.

COMSOL thermal modeling: In this thermal simulation, the initial temperature of the entire system was set to the ambient temperature of 293.15 K. The A , ε , k , ρ , and C_p of LIG were set as 0.977, 0.98, 129 W (m K)⁻¹, 2200 kg m⁻³, and 1150 J (kg K)⁻¹. The primary uncertainty in our thermal model stems from the precise thermophysical properties of the LIG composite. As a porous material, these properties can vary within a certain range based on the laser processing parameters and the distribution of HEA-NPs. While these variations introduce minor quantitative errors in the absolute temperature values predicted, they do not alter the overall conclusions of our study. The simulated trends, which include the rapid thermal response, cooling process and uniform temperature distribution, are in qualitative agreement with our experimental results. (Supplementary Note 2, supplementary information)

COMSOL Optical modeling: The electric displacement field is described using the refractive index model:

$$\varepsilon_r = (n - ik)^2 \quad (\text{S22})$$

where n is the refractive index, and k is the extinction coefficient. The structure parameters for simulation were based on experimental CLSM profiles. The plane wave source with a

wavelength from 3 to 20 μm (with a spacing of 0.05 μm) was incident along the y axis and the periodic boundary condition (PBC) was applied to the surfaces along the x axis:

$$E_{dst} = E_{src} \quad (\text{S23})$$

$$H_{dst} = H_{src} \quad (\text{S24})$$

The incident boundary is set using the scattering boundary condition:

$$\mathbf{n} \times (\nabla \times \mathbf{E}) - jk\mathbf{n} \times (\mathbf{E} \times \mathbf{n}) = 0 \quad (\text{S25})$$

In addition, perfectly matched layer (PML) absorbing boundary conditions were applied in the y axis to the top and bottom of the air and materials layer, which effectively simulated an open free-space environment. Along the x axis, the width was set to 300 μm in all simulations. The n and k of air was set as 1 and 0. The n and k of Al model adopted the material parameters built-in in COMSOL software (Rakic 1998: Lorentz-Drude model; n,k 0.0620-248 μm). The n and k of graphene parts in the P-LIG and HEAs/LIG models adopted the material parameters built-in in COMSOL software (Query 1985: Pyrolytic carbon; n,k 0.21-55.6 μm). For the metal parts of HEAs/LIG, the n and k used the Fe as example, which also adopted the built-in parameters in COMSOL software (Query 1985: n,k 0.21-55.6 μm). For the optical modeling and the calculation of emissivity, the main sources of uncertainty are the n and k of the composite, as well as the simplification of modeling the metallic inclusions primarily based on Fe as a representative element. In addition, the minor contribution from the metal oxides, whose volume fraction was below the detection limit of XRD, was omitted. All of the factors could introduce a degree of approximation. Considering these, our model likely provided a lower-bound estimate of the performance of HEAs/LIG composite. While the absolute values from our optical simulations carried a degree of uncertainty, the key physical insights and the superior radiation performance of our HEA/LIG composite were demonstrated and are not compromised by these approximations. (Supplementary Note 8, supplementary information)

Reviewer's Comments

5. Given the local nature of femtosecond direct writing, the manuscript should address large-area uniformity, batch-to-batch reproducibility, and practical scalability/throughput considerations.

Our Response

We sincerely thank the reviewer for these critical questions regarding large-area uniformity, reproducibility, and scalability, which are essential for assessing the practical potential of our technology. We have conducted systematic evaluations to address each of these points, and the results demonstrate the robustness of our femtosecond laser direct writing process.

To demonstrate large-area uniformity, we fabricated a HEAs/LIG sample with an active area of 40 mm × 40 mm (**Figure R7a**). The infrared thermal image of this sample under an applied voltage of 16 V revealed a highly uniform temperature distribution (**Figure R7b**). This excellent uniformity is achieved by our femtosecond laser direct writing process, which involves point-by-point material transformation followed by a continuous line-scanning strategy. The overlapping of laser spots and scans ensures consistent energy delivery, leading to homogeneous formation of both the LIG substrates and HEA-NPs. However, fabricating even larger areas (e.g., beyond 100 mm × 100 mm) with our current lab-scale setup is limited by the scanning field of the galvanometer and the laser power. These are not fundamental limitations of the technology itself. Laser systems with large-field lenses and higher-power lasers will be available for industrial-scale processing.

We evaluated the batch-to-batch reproducibility by measuring the sheet resistance at multiple locations across five independent HEAs/LIG samples fabricated over a span of eight months (including samples from our earlier studies) (**Figure R8**). The results show a very narrow distribution in sheet resistance (**Figure R9**). This remarkably low variation displayed the exceptional batch-to-batch reproducibility of our method. The digital and programmable nature of laser writing ensures consistent energy delivery and patterning across different batches.

Regarding throughput, the processing time for our method is directly proportional to the area. For instance, fabricating a 20 mm × 20 mm sample takes ~20 minutes in our current configuration. This represents a highly efficient and single-step process that simultaneously synthesizes the LIG support and the HEA-NPs in ambient air. The current high uniformity and reproducibility are achieved using a Gaussian laser beam. However, the current single-beam

scanning speed may present a throughput challenge for very large-scale fabrication. The most promising route for industrial-scale throughput is parallelization, which uses beam shaping with a spatial light modulator (SLM) to create multiple focused spots or a homogeneous flat-top beam, allowing for the parallel writing of multiple lines or large areas in a single pass. This would dramatically increase the throughput and makes the process compatible with roll-to-roll manufacturing for flexible electronics. It is also a key focus of our ongoing research.

Figure R7. (a) Optical photo and (b) infrared camera image of a larger HEAs/LIG sample. The size of the as-prepared sample area was 40 mm × 40 mm.

Figure R8. Optical photos of HEAs/LIG prepared at different time point in 2025.

Figure R9. Sheet resistance of HEAs/LIG prepared at different time point in 2025.

We have added the discussion about the large-area uniformity, batch-to-batch reproducibility, and practical throughput in the revised manuscript. We also included **Figure R7** as **Figure S12**, **Figure R8** as **Figure S13**, and **Figure R9** as **Figure S14** in the supplementary information, as follows:

We also fabricated a HEAs/LIG sample with an active area of 40 mm × 40 mm, and the infrared thermal image of this sample under an applied voltage revealed a highly uniform temperature distribution (Supplementary Fig. 12). This excellent uniformity is achieved by our femtosecond laser direct writing process, which involves point-by-point material transformation followed by a continuous line-scanning strategy. Laser systems with large-field lenses and higher-power lasers will be available for industrial-scale processing. We also evaluated the batch-to-batch reproducibility by measuring the sheet resistance at multiple locations across five independent HEAs/LIG samples fabricated over a span of eight month (Supplementary Fig. 13). The results show a very narrow distribution in sheet resistance (Supplementary Fig. 14). This remarkably low variation displayed the exceptional batch-to-batch reproducibility of our method. The processing time for our method preparing HEAs/LIG samples is directly proportional to the area. This represents a highly efficient and single-step process that simultaneously synthesizes the HEAs/LIG in ambient air. The most promising route for industrial-scale throughput is parallelization, which uses beam shaping with a spatial light modulator (SLM) to create multiple focused spots or a homogeneous flat-top beam, allowing for the parallel writing of multiple lines or large areas in a single pass. (Page 5, Manuscript)

Reviewer's Comments

6. Bending/adhesion/abrasion durability data are important for flexible device claims and should be presented with clear before/after performance metrics.

Our Response

We sincerely thank the reviewer for this valuable comment regarding the durability of flexible HEAs/LIG electric heater. Following your suggestion, we have conducted a series of

standardized mechanical tests, including bending, adhesion, and abrasion tests, to quantitatively evaluate the durability.

To evaluate the flexibility, the HEAs/LIG sample was subjected to 200 repeated bending cycles (Figure R10a). The equilibrium temperature was recorded during the bending test (Figure R10b). Figure R10c further shows the temperature-time curves of the HEAs/LIG heater at a constant voltage of 8 V before and after 200 bending cycles. The saturation temperature remained nearly unchanged. This excellent bending stability demonstrated that the HEA-NPs were firmly anchored on the flexible LIG network. The LIG substrate itself could accommodate strain without fracture, and our fabrication method ensured that the composite structure maintains its electrothermal performance under repeated mechanical deformation. This result supports the flexibility for applications involving bending.

We further assessed the mechanical robustness including adhesion and abrasion, which are critical for skin-wearable or frequently handled devices. A standard tape test using a 3M transparent tape was performed for 30 cycles. The tape was firmly pressed onto the HEAs/LIG surface and then peeled off (Figure R11). The sample was subjected to a linear abrasion test over a total path length of 10 cm under a normal load of 100 g with 800 mesh sand paper (Figure R12). Every each test was recorded as one cycle. The results indicated that these tests led to a noticeable performance degradation. The equilibrium temperature decreased from $\sim 202\text{ }^{\circ}\text{C}$ to $\sim 29\text{ }^{\circ}\text{C}$. The current HEAs/LIG structure exhibits limited resistance to mechanical wear and strong adhesion forces. The performance is owing to the physical removal of the top-layer LIG flakes and HEA-NPs. For applications requiring adhesion and abrasion resistance, encapsulation with a thin and flexible polymer layer, such as Polydimethylsiloxane (PDMS) and Ecoflex, would be a highly effective solution.

Figure R10. (a) Optical photo, (b) equilibrium temperature change, and (c) temperature-time curves of bending test for the HEAs/LIG over 200 cycles at the voltages of 8 V.

Figure R11. (a) Optical photo of the HEAs/LIG after 30 adhesion cycles. (b) Equilibrium temperature change of the HEAs/LIG over 30 cycles at the voltages of 8 V.

Figure R12. (a) Illustration of one abrasion cycle. (b) Optical photo of the HEAs/LIG after 5 abrasion cycles. (c) Equilibrium temperature change of the HEAs/LIG over 5 cycles at the voltages of 8 V.

We have added the discussion about the bending durability in the revised manuscript. We also included **Figure R10** as **Figure S53**, in the supplementary information, as follows:

The bending stability of HEAs/LIG heater was examined. We performed 200 cycles bending tests and monitored the evolution of its electrothermal performance metrics (Supplementary

Fig. 53). The electrothermal response remained unchanged after testing. This exceptional durability under mechanical deformation originated from the synergistic combination of the flexible LIG substrate and the anchored HEA-NPs. (Page 12, Manuscript)

Reviewer's Comments

7. If energy-efficiency advantages over commercial heaters are asserted, the underlying assumptions, parameters, and calculation approach should be transparent.

Our Response

We thank the reviewer for this rigorous attention to the calculation details of our energy-efficiency framework. We fully agree that claims of energy efficiency must be supported by transparent calculations. Our claim of energy-efficiency advantage is primarily based on a lower steady-state power consumption for achieving a comparable surface temperature. We have revised the manuscript to include a dedicated section (Supplementary Note 9, supplementary information) detailing the following assumptions, parameters, and methodology. The revisions can be found in the revised manuscript and supplementary information:

The detailed assumptions, parameters, and methodology of the energy saving calculation could be found in Supplementary Note 9. Using the computational models, we mainly calculated the winter energy savings potential across China. Compared to using CEH, HEAs/LIG heater delivers markedly higher energy saving performance. For example, Harbin can save roughly 1134.99 MJ m⁻². (Page 15, Manuscript)

A comprehensive Python-based simulation framework was used to evaluate the winter energy-saving performance of HEAs/LIG as a Joule heater compared with CEH. First, we set several common parameters. The climate data used includes the dataset for mean air temperature over the main terrestrial lands of China 2020. At the same time, building density and roof area ratios were set based on statistical data from various provinces in China. For example, Beijing's building density was set at 0.35, and roof area accounted for 30% of the building land area. The energy-efficiency analysis was based on the experimental data in Fig. 5i. When raising the ambient temperature from -7.5 °C to the winter indoor comfort level of 24 °C, the required

power for the HEAs/LIG and CEH are 1.17 W and 2.30 W, respectively, over the same area. Using 24 °C as the target indoor temperature, power density scaling was performed based on the above temperature difference, where the scaling factor equals the heating demand divided by the baseline temperature difference. This yielded the difference in power consumption between HEAs/LIG and the CEH when used as electric heating devices. Assuming continuous operation over 24 hours without considering differences in building insulation, the total difference in energy savings density over a 90-day period was ultimately determined. (Supplementary Note 9, supplementary information)

Reviewer's Comments

8. The use of Ru raises questions about cost and supply. A brief discussion of material criticality and potential substitutions would improve the practicality narrative.

Our Response

We sincerely thank the reviewer for this insightful comment regarding the cost and supply risk of Ru. Ru is a platinum-group metal typically obtained as a by-product. In our initial design, Ru was selected primarily to modestly increase configurational entropy rather than to leverage any Ru-specific noble-metal functionality. However, we fully acknowledge the practical limitations of Ru, as rightly pointed out by the reviewer. To address this, we have already identified promising substitution strategies by referring literatures.

Cu is a highly viable candidate, as it is inexpensive and possesses excellent electrical conductivity. In addition to extending the versatility of the femtosecond laser synthesis for HEA-NPs, we will aim to leverage higher conductivity for achieving uniform heating and rapid thermal response. This can also ensure high performance while enhancing practicality and scalability. Furthermore, we will also screen Zn, Mo, and Sn as alternative or additional elements to prepare single-phase HEAs while avoiding cost and supply risk. It is worth mentioning that our femtosecond laser process forms HEA-NPs directly on the LIG, the areal loading of every element (Fe, Co, Ni, Cr, Mn, and Ru) is very low, with an expected total

loading density of $\sim 1.5 \text{ mg cm}^{-2}$. Therefore, even if Ru was used, its contribution to device-level cost is limited.

To improve the practicality narrative of HEAs/LIG Joule heater, we have added the discussion about the material criticality and potential substitutions in the revised manuscript:

Based on the concentration of the FeCoNiCrMnRu precursor solution (0.1 M), the used volume (160 μL) and the coated area (20 mm \times 20 mm), we can estimate the mass loading of the FeCoNiCrMnRu HEA-NPs on the LIG to be $\sim 1.5 \text{ mg cm}^{-2}$. Therefore, even if Ru was used with considering its cost and supply, its contribution to device-level cost is limited. However, it is also desirable to investigate a wider range of low-cost and abundant elements, such as Cu, Zn, Mo, and Sn, to either replace Ru or further increase the configurational entropy, which is anticipated to correspondingly enhance the electrothermal performance of the HEAs/LIG composite. (Page 5, Manuscript)

Reviewer's Comments

9. A direct comparison between FeCoNiCrMn (high-entropy) and FeCoNiCrMnRu is needed to reveal whether performance gains arise from Ru itself or from marginally higher configurational entropy.

Our Response

We sincerely thank the reviewer for this insightful comment regarding the role of Ru. To verify the significance of Ru and configurational entropy (ΔS_{conf}), we fabricated and tested three HEAs/LIG composites: FeCoNiCrMnRu (with Ru and high ΔS_{conf}), FeCoNiCrRu (low ΔS_{conf}), and FeCoNiCrMn (no Ru) (**Figure R13**). The comprehensive results were presented in **Figure R14** (sheet resistance), **Figure R15** (Joule heating performance), **Figure R16** (infrared camera images), and **Figure R17** (infrared emissivity). The binary FeCoNiCrMnRu sample ($\sim 51.8 \Omega \text{ sq}^{-1}$) also showed a slightly lower sheet resistance compared to both quinary FeCoNiCrRu ($\sim 60.2 \Omega \text{ sq}^{-1}$) and FeCoNiCrMn ($\sim 59.9 \Omega \text{ sq}^{-1}$). Furthermore, the FeCoNiCrMnRu showed a performance improvement in equilibrium temperatures at the 8V voltage compared to both FeCoNiCrRu and FeCoNiCrMn. The equilibrium temperatures of FeCoNiCrMnRu,

FeCoNiCrRu, and FeCoNiCrMn were ~ 204.1 °C, ~ 183.4 °C, and ~ 185.3 °C, respectively. The infrared emissivity was consistently high across all three samples, indicating that this property is predominantly governed by the LIG carrier rather than the specific HEA composition. These results suggested that the metallic elements in this system contributed in a largely equivalent manner to the electrical and thermal properties. The slight performance advantage of the senary FeCoNiCrMnRu was more accurately attributed to its higher configurational entropy rather than to the specific chemical identity of Ru.

Figure R13. Optical photos of (a) FeCoNiCrMnRu HEAs/LIG, (b) FeCoNiCrRu HEAs/LIG, and (c) FeCoNiCrMn HEAs/LIG surfaces.

Figure R14. Sheet resistance of FeCoNiCrMn the HEAs/LIG, FeCoNiCrRu HEAs/LIG, and FeCoNiCrMnRu HEAs/LIG.

Figure R15. Temperature-time curves of the FeCoNiCrMn HEAs/LIG, FeCoNiCrRu HEAs/LIG, and FeCoNiCrMnRu HEAs/LIG at the voltages of 8 V.

Figure R16. Corresponding infrared camera images of the FeCoNiCrMn HEAs/LIG, FeCoNiCrRu HEAs/LIG, and FeCoNiCrMnRu HEAs/LIG at the voltages of 8 V.

Figure R17. Infrared emissivity of the FeCoNiCrMn HEAs/LIG, FeCoNiCrRu HEAs/LIG, and FeCoNiCrMnRu HEAs/LIG in the wavelength range of 2.5 to 20 μm .

We have revised the manuscript to incorporate this new data and the refined discussion. We also included **Figure R13** as **Figure S37**, **Figure R14** as **Figure S38**, **Figure R15** as **Figure**

S50, Figure R16 as Figure S51, and Figure R17 as Figure S56 in the supplementary information.

The senary FeCoNiCrMnRu sample also showed a slightly lower sheet resistance compared to both quinary FeCoNiCrRu ($\sim 60.2 \Omega \text{ sq}^{-1}$) and FeCoNiCrMn ($\sim 59.9 \Omega \text{ sq}^{-1}$) (Supplementary Figs. 37 and 38). (Page 9, Manuscript)

As for the high entropy samples, the FeCoNiCrMnRu showed a performance improvement in equilibrium temperatures at the 8V voltage compared to both FeCoNiCrRu and FeCoNiCrMn. Specifically, the equilibrium temperatures of the FeCoNiCrMnRu, FeCoNiCrRu, and FeCoNiCrMn were $\sim 204.1^\circ\text{C}$, $\sim 183.4^\circ\text{C}$, and $\sim 185.3^\circ\text{C}$, respectively (Supplementary Figs. 50 and 51). (Page 12, Manuscript)

The infrared emissivity was consistently high across all the FeCoNiCrMnRu, FeCoNiCrRu, and FeCoNiCrMn samples, further indicating that this property is predominantly governed by the LIG carrier rather than the specific HEA composition. (Supplementary Fig. 56). (Page 13, Manuscript)

Reviewer's Comments

10. For Joule-heating use cases, rapid on/off cycling stability should be demonstrated, with tracking of sheet resistance, heating metrics, and emissivity over extended cycles.

Our Response

We greatly appreciate the reviewer's insightful and constructive suggestion. We fully agree that the cycling stability is a critical performance metric for practical Joule-heating applications. We have conducted additional experiments to systematically evaluate the rapid on/off cycling stability of our HEAs/LIG heater. The HEAs/LIG sample was subjected to 100 consecutive rapid heating-cooling cycles under a constant DC voltage of 8 V. Each cycle consisted of 60 seconds of heating followed by 60 seconds of natural cooling in ambient conditions. During this long-term cycling test, we tracked the sheet resistance, equilibrium temperature, and infrared emissivity (**Figure R18**). The HEAs/LIG heater exhibits exceptional operational stability. The sheet resistance remained remarkably stable, which indicated that the conductive LIG network and the anchored HEA-NPs maintained excellent structural and electrical

integrity without significant oxidation, cracking, or delamination. Correspondingly, the equilibrium temperature also showed negligible degradation. The infrared emissivity of the sample remained constant throughout the stability test, confirming the surface chemical and morphological stability, which is crucial for radiation-based heating applications. The outstanding cycling stability can be attributed to the robust interface between the HEA-NPs and the LIG substrate, as well as the intrinsic high-temperature stability and anti-oxidation properties of the graphene shell-encapsulated HEA-NPs composition.

Figure R18. (a) Temperature-time curve of HEAs/LIG sample during the rapid on/off cycling stability test at the voltage of 8 V. (b) Evolution of sheet resistance over 100 cycles. (c) Equilibrium temperature change over 100 cycles. (d) Infrared emissivity of the heater at the initial, 20th, 40th, 60th, 80th, and 100th cycles in the wavelength range of 2.5 to 20 μm. Each complete heating/cooling cycle lasted for ~60 s.

We have added these new results into the revised manuscript and included **Figure R18** as **Figure S71** in the supplementary information.

It was worth noting that the HEAs/LIG heater showed excellent stability during the rapid on/off cycling test (Supplementary Fig. 71). The outstanding cycling stability comes from the robust

interface between the HEA-NPs and the LIG substrate, as well as the intrinsic high-temperature stability and anti-oxidation properties of the graphene shell-encapsulated HEA-NPs composition. (Page 15, Manuscript)

REVIEWER #2

Reviewer's Comments

This manuscript presents the synthesis of high-entropy alloy nanoparticles (HEA-NPs) on laser-induced graphene (LIG) using femtosecond (fs) laser irradiation, aiming to demonstrate enhanced Joule heating performance. However, this reviewer finds several fundamental concerns that undermine the validity and significance of the reported results.

Our Response

We are grateful to the reviewer for their careful feedback on our manuscript. Your comments are very insightful and helpful to improve our manuscript. Below are our detailed responses to each raised concern and suggestion.

Reviewer's Comments

1. The justification for using a femtosecond laser is insufficient. While the authors claim a thermal mechanism for HEA nanoparticle formation, femtosecond lasers are typically associated with non-thermal, "cold" ablation processes. The manuscript does not demonstrate clear advantages of fs lasers over more accessible alternatives like nanosecond or CW lasers, raising concerns about the method's practicality and relevance.

Our Response

We sincerely thank the reviewer for this valuable comment regarding the practicality and relevance of femtosecond laser synthesis HEA-NPs. Here is our point-to-point response.

The initial interaction of a single femtosecond laser pulse with a material is often non-thermal, dominated by direct field ionization and Coulomb explosion. This "cold" ablation process is indeed beneficial for high-precision machining. It should be noted that this process often usually occurs at the low repetition rate regime, where thermal accumulation can be usually negligible. The surface peak temperature decreases to the initial degree before the next

laser pulse arrives. These features minimize the thermal collateral damage and heat affected zone. In this regime, the repetition rate is usually a few kHz. However, the fabrication efficiency is also limited by the low pulse repetition rate [*Laser Photonics Rev.*, 2021, 15, 2000455]. When the repetition rate of femtosecond laser pulse reaches several hundred kHz (typically above 100 kHz), the time interval between consecutive laser pulses becomes shorter than the duration required for the absorbed energy to diffuse out of the focal volume. This will lead to significant local thermal accumulation effects. While this conclusion was originally drawn from studies on femtosecond laser processing of glass substrates, in our work, we utilized the LIG, which exhibits ultrahigh light absorptivity, as the substrate and selected a high femtosecond laser repetition rate of 400 kHz. This configuration offers distinct advantages for the synthesis of HEA-NPs. The thermal formation mechanism of HEA-NPs has been validated by many research [*Science*, 2022, 376, eabn3103; *Adv. Mater.*, 2025, 37, 2412337]. Following the ultrafast heating induced the simultaneous decomposition of all metal salts, the molten droplet undergoes an ultrafast quenching process. This extreme quenching rate is an important factor that kinetically traps the multiple immiscible metallic elements into a single-phase HEA solid solution, suppressing the phase separation and nucleation of intermetallic compounds that would occur under slower cooling. Therefore, the femtosecond laser creates a unique ultrafast thermal cycle that is perfectly suited for HEA formation. These advantages of the femtosecond laser are reflected in the synthesized HEA-NPs.

Based on our experimental findings and supporting literature, the femtosecond laser demonstrates obvious advantages over nanosecond and continuous wave (CW) lasers for the synthesis of HEA-NPs. Femtosecond laser synthesis enables the formation of significantly smaller HEA-NPs compared to those typically obtained with CW laser [*J. Mater. Chem. A*, 2024, 12, 21744-21757]. This is a direct consequence of the ultrahigh heating and cooling rates, which rapidly freeze the nucleated liquid droplets, effectively suppressing coalescence that lead to coarse and irregular particles under the slower thermal cycles of CW laser. A key practical advantage is the ability to synthesize HEA-NPs directly in ambient atmosphere without requiring liquid environments or chemical reductants. Many nanosecond laser methods or other techniques for producing non-oxide HEA-NPs necessitate such controlled conditions [*Adv. Mater.*, 2025, 37, 2504099; *Light Sci. Appl.*, 2024, 13, 270; *Energy Environ. Sci.*, 2024, 17,

8670-8682; *Adv. Funct. Mater.*, 2023, 33, 2211279; *Nat. Synth.*, 2022, 1, 138-146; *Adv. Funct. Mater.*, 2022, 32, 2110645]. The femtosecond laser achieves this through ultrafast and non-equilibrium energy deposition, while the extremely short interaction time minimizes oxidation of HEA-NPs despite being performed in air. Furthermore, the femtosecond laser induced multi-layer graphene-encapsulated HEA-NPs structures contribute to enhancing the durability of the electrothermal device and effectively prevents subsequent oxidation of the HEA-NPs, as verified by experiments [*Adv. Mater.*, 2024, 36, 2402391].

To improve the practicality and relevance of femtosecond laser synthesis HEA-NPs, we have added and corrected this crucial discussion about the advantages of femtosecond laser in the revised manuscript and supplementary information:

The femtosecond laser demonstrates obvious advantages over nanosecond and continuous wave (CW) lasers for the synthesis of HEA-NPs (Supplementary Table 6). Femtosecond laser synthesis enables the formation of significantly smaller HEA-NPs compared to those typically obtained with CW laser⁴⁰. This is a direct consequence of the ultrahigh heating and cooling rates, which rapidly freeze the nucleated liquid droplets, effectively suppressing coalescence that leads to coarse and irregular particles under the slower thermal cycles of CW laser. A key practical advantage is the ability to synthesize HEA-NPs directly in ambient atmosphere without requiring liquid environments or chemical reductants. Many nanosecond laser methods or other techniques for producing non-oxide HEA-NPs necessitate such controlled conditions^{32,41-46}. The femtosecond laser achieves this through ultrafast and non-equilibrium energy deposition, while the extremely short interaction time minimizes oxidation of HEA-NPs despite being performed in air. (Page 4, manuscript)

Supplementary Table 6. Comparative analysis of laser synthesis methods for HEAs.

Laser source	Supporter	Atmosphere	Reductant	Production	Ref.
Femtosecond laser	LIG	Air	None	HEA-NPs	Our work
Nanosecond laser	None	Liquid	None	Au-based HEA-NPs	35

Nanosecond laser	MXene	Ar	None	HEA-NPs	21
Nanosecond laser	LIG	Ar	None	HEA-NPs	22
CW laser	CP	Air	Sodium citrate	HEA-NPs	20
CW laser	None	Liquid	None	HEAs	36
CW laser	CNTs	N ₂	None	HEA-NPs	19
Nanosecond laser	Various	Liquid	None	HEA-NPs	37
Nanosecond laser	MWCNTs	N ₂	None	HEA-NPs	38

Reviewer's Comments

2. The thermal simulation based on the classical heat diffusion equation is also problematic. It fails to capture ultrafast non-equilibrium dynamics occurring in the first few picoseconds after fs-laser exposure, leading to questionable predictions such as a ~ 3000 K peak temperature and a cooling rate of $\sim 5.6 \times 10^{11}$ K/s. Figures 1b–d rely heavily on this model, without addressing material phase stability or the validity of such extreme thermal conditions. Key physical effects such as laser ablation, evident in Fig. S13, and heat accumulation from pulse overlap are neglected in the simulation. These omissions limit the credibility of the thermal analysis and the claimed mechanism of HEA nanoparticle formation.

Our Response

We sincerely thank the reviewer for these insightful and critical comments regarding our thermal simulation. Below are our point-to-point response.

We agree with the reviewer that the classical thermal diffusion model has limitations, particularly in its inability to capture the ultrafast non-equilibrium dynamics between electrons

and the lattice within the first few picoseconds following femtosecond laser exposure. A more rigorous approach for describing such ultrafast phenomena is indeed the two-temperature model (TTM), which has been extensively validated for confirmed metal or semiconductor related systems and can effectively simulate the temporal evolution of both electron and lattice temperatures [*Adv. Mater.*, 2025, e12727; *Nano-Micro Lett.*, 2025, 17, 179; *Adv. Funct. Mater.*, 2025, 35, 2424526]. However, the application of the TTM requires a set of precise parameters, such as the electron-phonon coupling factor, electronic heat capacity, and electron thermal conductivity. In our study, the irradiated material is a complex composite of pyrolytic porous graphene and high-entropy metallic precursors. For our specific composite system, the key parameters required for TTM are not well-defined and established at this stage. Considering this constraint, we therefore adopted the classical heat diffusion model as a simplified but tractable alternative to provide qualitative insights into the thermal profiles. While we recognize that this classical heat diffusion model is coarse and ignores some nuances of ultrafast dynamics, it has been effectively employed in prior femtosecond laser studies on carbon-based materials to explain rapid thermal cycles [*ACS Nano*, 2023, 17, 18893-18904; *Adv. Funct. Mater.*, 2023, 33, 2213514; *Adv. Opt. Mater.*, 2021, 9, 2100793]. It effectively provides a qualitative understanding for the trend of ultrahigh heating and cooling rates, such as the peak temperature of ~ 3000 K and cooling rate of $\sim 5.6 \times 10^{11}$ K s⁻¹. The ultrahigh peak temperature is indicative of a regime where the absorbed energy is sufficient to cause precursors to decompose simultaneously, which is a necessary condition for the formation of HEA-NPs. The ultrahigh cooling rate is a fundamental and well-established principle for suppressing phase segregation and promoting the stabilization of a single solid-solution phase. Thus, while the exact numerical value is determined by the model, the phenomenon of ultrafast heating/cooling and its consequence is physically reasonable, which is supported by many existing research [*Science*, 2022, 376, eabn3103; *Nat. Synth.*, 2022, 1, 138-146; *Nat. Commun.*, 2025, 16, 3403; *Light Sci. Appl.*, 2024, 13, 270; *Adv. Mater.*, 2025, 37, 2412337]. These extreme thermal conditions are essential for HEA-NPs formation. The high entropy stabilizes the solid-solution at high temperatures, and the rapid cooling kinetically "freezes" this state. The experimental characterizations of the resulting particles also support the FeCoNiCrMnRu

HEA-NPs formation. Therefore, the predicted extreme thermal conditions are not only plausible but are the driver for the observed HEA nanoparticle formation (**Figure R19**).

We agree that our model simplifies some important physical effects such as neglecting laser ablation and heat accumulation from pulse overlap. The heat accumulation from pulse overlap is also a critical factor for the overall processing outcome. In this initial study, our simulation was focused on the thermal response induced by a single femtosecond laser pulse to understand the fundamental thermal formation mechanism of HEA-NPs from the precursor material. Additionally, the high heat dissipation efficiency of the graphene substrate helps minimize potential damage to the substrate caused by heat accumulation effects under high repetition rate femtosecond laser. The simplification is an effective approach to decouple the complex multi-pulse effects. We fully know that the model can be refined, and we are actively working on developing multi-pulse models, which can fully describe the macroscopic morphology.

We appreciate the professional advice provided by the reviewers. We will clarify the limitations and the qualitative nature of our thermal analysis in the revised manuscript. Further refinement of the thermal model and potentially through experimental determination of the necessary parameters for this model will be a key focus of our future work.

Figure R19. HAADF-STEM images and the corresponding EDS elemental mappings for HEA-NPs (Fe, Co, Ni, Cr, Mn, and Ru).

We have revised the manuscript to include a more explicit discussion of these model limitations and the interpretive nature of the absolute values, ensuring a balanced presentation of our thermal analysis:

The highly localized femtosecond laser pulse could facilitate rapid cooling and help preserve the single-phase random atomic configuration to promote the formation of uniform HEA solid solutions^{3,9,10}. (Page 4, manuscript)

It is noted that the classical heat diffusion model employed here provides an estimate of the extreme thermal cycle but does not capture the initial electron-lattice non-equilibrium. The predicted temperatures and heating/cooling rates should be interpreted as the qualitative

conditions of the extreme conditions that facilitate rapid melting and solidification, consistent with the formation of HEA-NPs. While the exact numerical value is determined by the model, the phenomenon of ultrafast heating/cooling and its consequence is physically reasonable. (Page 4, manuscript)

Reviewer's Comments

3. Lastly, the metric of “electrothermal efficiency” appears equivalent to area-normalized thermal resistance, which may overstate performance for thinner films. The comparison in Fig. 4g seems not fair, as many cited devices were designed for multifunctionality, not optimized heating.

Our Response

We thank the reviewer for this astute observation and raising these important points regarding the electrothermal performance metrics and comparison. We fully agree with the reviewer that a fair comparison is crucial. As for the electric heater, upon reaching a steady temperature, a power balance is established between the input power and the power losses due to thermal radiation and convection:

$$\frac{U_0^2}{R} = h_c A (T_s - T_0) + h_R A (T_s^4 - T_0^4)$$

where U_0 is the applied voltage, R is the resistance, h_c is the convective heat transfer coefficient, A is the working surface area, T_s is the stable temperature, T_0 is the ambient temperature, h_R is the radiative heat transfer coefficient, respectively. In fact, our defined electrothermal efficiency (E_T), which is the slope of the fitted line from input power density to saturated temperature), is the inverse of the total heat transfer coefficient:

$$E_T = \frac{1}{h_c + h_R}$$

when the E_T is higher, less power density is required for reaching the same saturated temperature. E_T quantifies how efficiently electrical power is converted into a temperature increase against environmental heat losses. We deliberately chose this normalized parameter ($^{\circ}\text{C cm}^2 \text{ W}^{-1}$) to enable a cross-comparison of different electrothermal materials on a relative

performance basis, as it effectively eliminates the influence of the heated area. This electrothermal efficiency metric is somewhat different from the area-normalized thermal resistance, which also takes into account radiative heat dissipation. The E_T is governed by convection and radiation to the ambient, which is insensitive to the film thickness.

The comparison in **Figure R20** aims to situate our work within the broader field of the thin-film heaters and flexible thermal management platforms. We agree that some of the cited devices were designed with additional functionalities. However, a primary function highlighted and evaluated in all these cited works is their performance as electrothermal heaters. The comparison is made against their reported heating capabilities that used the same or similar metrics. All devices included in the comparison are thin film heaters, ensuring a relevant and fair comparison of their fundamental electrothermal properties. From another perspective, our prepared HEAs/LIG is also a multifunctional platform. In our work, the HEAs/LIG composite is designed for Joule heating, but it might also possess significant potential for electrocatalysis, sensing, and other applications [*Light Sci. Appl.*, 2024, 13, 270]. The excellent electrothermal performance reported here is a valuable functionality achieved on this versatile platform. Finally, the reviewer raises an excellent point about multifunctionality. Indeed, while the primary focus of our work currently focuses on ultrafast-laser synthesis of HEA-NPs on LIG as a high-efficiency Joule heater, we are actively investigating broader functionalities of HEAs/LIG.

Figure R20. Comparison of electrothermal conversion efficiency between the HEAs/LIG and various heaters.

We have also added corresponding discussion about the multifunctionality of **Figure 4g** in the revised manuscript:

This comparison was intended to provide a performance standard for our prepared HEAs/LIG within the landscape of reported advanced film and flexible electrothermal devices. It was acknowledged that some of the compared devices were developed for multifunctional applications. (Page 12, Manuscript)

REVIEWER #3

Reviewer's Comments

In this manuscript, they achieved the successful fabrication of FeCoNiCrMnRu HEA-NPs on LIG by using femtosecond laser ultrafast solid-phase synthesis. It is a novel research. Some parts should be revised. It can be accepted after revisions.

Our Response

We appreciate the reviewer's valuable comments and your recognition of the novelty and scientific merits of this manuscript. Below are our detailed responses to each raised concern and suggestion.

Reviewer's Comments

1. Pristine LIG (P-LIG) was prepared using femtosecond laser direct writing technology. How much of the yield of the nanoparticles should be menthened? How many grams did you prepared?

Our Response

We sincerely thank the reviewer for this insightful question. The FeCoNiCrMnRu HEA-NPs in this work were synthesized in situ and are strongly anchored to the fibrillar network of the P-LIG substrate and used as the Joule heater. They are not collected as a free-standing powder. Therefore, the metrics of yield and grams prepared are not directly applicable to our laser synthesis process. However, based on the concentration of the FeCoNiCrMnRu precursor solution (0.1 M), the used volume (160 μ L) and the coated area (20 mm \times 20 mm), we can estimate the mass loading of the FeCoNiCrMnRu HEA-NPs on the LIG to be ~ 1.5 mg cm⁻².

As the HEA-NPs are not independent, a precise mass in grams cannot be provided. However, we can describe the scale of our preparation. In a typical experiment, we prepared a HEAs/LIG sample with a geometric area of 20 mm \times 20 mm covered with the HEA-NPs

composites. Using the mass loading value above for a typical sample, the total mass of HEA-NPs synthesized per sample was ~6.0 mg. This entire preparation process took less than 20 minutes and exhibited high efficiency.

Reviewer's Comments

2. How to control the particle size to make it uniform by so fast laser processing. A TEM image of the nano HEA particles should be added.

Our Response

We sincerely thank and appreciate for this insightful comment regarding the control and uniformity of the HEA-NPs size. We have conducted additional experiments to investigate the effect of laser scanning speeds. Specifically, we have now included STEM images and corresponding particle size distribution analyses of the synthesized HEA-NPs fabricated at different laser scanning speeds (50, 70, and 90 mm s⁻¹) (**Figure R21**). Our results consistently demonstrate that increasing the laser scanning speed leads to a decrease in the average nanoparticle size [*Light Sci. Appl.*, 2024, 13, 270]. This trend can be attributed to the reduced interaction time and effective growth time for the nanoparticles at higher scanning speeds, which limits the coalescence. While a general trend is established, the current STEM images show some larger nanoparticles within the distribution. We attribute this primarily to the inherent temperature gradient generated by the Gaussian profile of the femtosecond laser beam. This profile creates a non-uniform thermal field where the center of the laser spot reaches a higher temperature, potentially leading to increased coalescence and the formation of a minority of larger nanoparticles. To address this and achieve superior size control with a more uniform distribution in future work, we plan to employ laser beam shaping techniques to create a more uniform energy distribution across the processing area. This is expected to minimize localized hot spots and yield nanoparticles with a more uniform size distribution.

Figure R21. STEM images and particle size distribution analysis of HEAs/LIG with laser scanning speeds of (a) 50, (b) 70, and (c) 90 mm s⁻¹.

we have also added the above **Figure R21** as **Figure S36** in the supplementary information and discussion about the particle size in the revised manuscript:

Furthermore, the particle size of the FeCoNiCrMnRu HEA-NPs was found to decrease with increasing laser scanning speed (Supplementary Fig. 36). This trend can be attributed to the reduced interaction time and effective growth time for the nanoparticles at higher scanning speeds, which limits the coalescence. (Page 8, Manuscript)

Reviewer's Comments

3. How can you find it for high-performance Joule heating applications? occasionally?

Our Response

We thank the reviewer for this insightful question regarding the high-performance Joule heating applications of HEAs/LIG. Our research originally focused on the electrical conductivity and electrothermal conversion performance of LIG. During these research, we recognized that while the LIG holds great promise, there was an need in this field to further

enhance its electrical and thermal properties for more demanding applications. This directed our attention to the emerging field of HEAs.

Traditionally, there existed performance tradeoffs where high-conductivity metals possess low emissivity, while the metal oxides with high emissivity often had poor electrical conductivity. To overcome this, we devised a synergistic design strategy (**Figure R22**). From the perspective of electrical conductivity, the HEA-NPs acted as the conductive bridges within the porous network of LIG. This effectively lowered the overall sheet resistance of the composite film (from 90.7 to 51.8 $\Omega \text{ sq}^{-1}$). From the perspective of infrared emissivity, we leveraged the intrinsic 3D porous and interconnected structure of LIG, which is highly effective at trapping and emitting infrared radiation. This design ensures that the composite still maintained a high broadband infrared emissivity of ~ 0.98 across a broad wavelength range of 2.5–20 μm despite the incorporation of HEA-NPs, which might exhibit lower emissivity.

The key to realizing this design was our ultrafast femtosecond laser synthesis method. We discovered that this method could directly synthesize HEA-NPs on the LIG in ambient air. This one-step, in-situ fabrication method proved to be highly effective, which created ideal non-equilibrium conditions for HEA-NPs formation. Consequently, the resulting HEAs-decorated LIG composites demonstrated a significant enhancement in Joule heating, alongside outstanding electrothermal conversion efficiency ($\sim 285.4 \text{ }^\circ\text{C cm}^2 \text{ W}^{-1}$) and cycle stability. The overall performance surpassed that of most previously reported materials, as presented in the manuscript.

Figure R22. Schematic illustration of the design of high-performance Joule heating materials enabled by HEA-NPs and LIG.

Reviewer's Comments

4. Why did you choose FeCoNiCrMnRu? Ru is very expensive? Why must you add Ru? How about the property without Ru?

Our Response

We greatly appreciate the reviewer's valuable questions. The element selection, cost-effectiveness of Ru, and its necessity are crucial for the material design strategy. We have conducted further experiments and refined our discussion to provide a more comprehensive and accurate explanation. Our selection of FeCoNiCrMn was primarily based on literature survey and considerations of configurational entropy (**Figure R23**), because it represents a well-established, foundational HEA system with proven phase stability [*Nat. Commun.*, 2022, 113, 2662; *Adv. Sci.*, 2023, 10, 2300426; *Energy Environ. Sci.*, 2024, 17, 8670-8682]. Building upon this, we further investigated the possibility of enhancing the configurational entropy by incorporating Ru to form FeCoNiCrMnRu. This approach has been experimentally verified as feasible, and the resulting slight increase in configurational entropy is expected to be beneficial for electrothermal conversion performance.

We agree that the Ru is relatively expensive and its supply can be constrained. In our initial design, Ru was selected primarily to modestly increase configurational entropy rather than to leverage any Ru-specific noble-metal functionality. However, our femtosecond laser process forms HEA-NPs directly on the LIG, the areal loading of every element (Fe, Co, Ni, Cr, Mn, and Ru) is very low, with an expected total loading density of $\sim 1.5 \text{ mg cm}^{-2}$. Therefore, even if Ru was used, its contribution to device-level cost is limited. Indeed, Ru is not indispensable. It was included as one option to lift configurational entropy with a chemically robust element. Other metals (e.g., Cu, Mo, Zn, Sn) can substitute this role in future iterations with similar design logic for preparing single-phase HEAs while avoiding cost and supply risk.

To verify the significance of Ru and configurational entropy (ΔS_{conf}), we fabricated and tested three HEAs/LIG composites: FeCoNiCrMnRu (with Ru and high ΔS_{conf}), FeCoNiCrRu (low ΔS_{conf}), and FeCoNiCrMn (no Ru) (**Figure R24**). The comprehensive results were

presented in **Figure R25** (sheet resistance), **Figure R26** (Joule heating performance), **Figure R27** (infrared camera images), and **Figure R28** (infrared emissivity). The senary FeCoNiCrMnRu sample ($\sim 51.8 \Omega \text{ sq}^{-1}$) also showed a slightly lower sheet resistance compared to both quinary FeCoNiCrRu ($\sim 60.2 \Omega \text{ sq}^{-1}$) and FeCoNiCrMn ($\sim 59.9 \Omega \text{ sq}^{-1}$). Furthermore, the FeCoNiCrMnRu showed a performance improvement in equilibrium temperatures at the 8V voltage compared to both FeCoNiCrRu and FeCoNiCrMn. Specifically, the equilibrium temperatures of FeCoNiCrMnRu, FeCoNiCrRu, and FeCoNiCrMn were $\sim 204.1 \text{ }^\circ\text{C}$, $\sim 183.4^\circ\text{C}$, and $\sim 185.3^\circ\text{C}$, respectively. The infrared emissivity was consistently high across all three samples, indicating that this property is predominantly governed by the LIG carrier rather than the specific HEA composition. These results suggested that the metallic elements in this system contributed in a largely equivalent manner to the electrical and thermal properties. The slight performance advantage of the senary FeCoNiCrMnRu was more accurately attributed to its higher configurational entropy rather than to the specific chemical identity of Ru.

Figure R23. HAADF-STEM images and the corresponding EDS elemental mappings for HEA-NPs (Fe, Co, Ni, Cr, and Mn).

Figure R24. Optical photos of (a) FeCoNiCrMnRu HEAs/LIG, (b) FeCoNiCrRu HEAs/LIG, and (c) FeCoNiCrMn HEAs/LIG surfaces.

Figure R25. Sheet resistance of FeCoNiCrMn the HEAs/LIG, FeCoNiCrRu HEAs/LIG, and FeCoNiCrMnRu HEAs/LIG.

Figure R26. Temperature-time curves of the FeCoNiCrMn HEAs/LIG, FeCoNiCrRu HEAs/LIG, and FeCoNiCrMnRu HEAs/LIG at the voltages of 8 V.

Figure R27. Corresponding infrared camera images of the FeCoNiCrMn HEAs/LIG, FeCoNiCrRu HEAs/LIG, and FeCoNiCrMnRu HEAs/LIG at the voltages of 8 V.

Figure R28. Infrared emissivity of the FeCoNiCrMn HEAs/LIG, FeCoNiCrRu HEAs/LIG, and FeCoNiCrMnRu HEAs/LIG in the wavelength range of 2.5 to 20 μm .

We have revised the manuscript to incorporate this new data and the refined discussion. We also included **Figure R24** as **Figure S37**, **Figure R25** as **Figure S38**, **Figure R26** as **Figure S50**, **Figure R27** as **Figure S51**, and **Figure R28** as **Figure S56** in the supplementary information.

Based on the concentration of the FeCoNiCrMnRu precursor solution (0.1 M), the used volume (160 μL) and the coated area (20 mm \times 20 mm), we can estimate the mass loading of the FeCoNiCrMnRu HEA-NPs on the LIG to be $\sim 1.5 \text{ mg cm}^{-2}$. Therefore, even if Ru was used with considering its cost and supply, its contribution to device-level cost is limited. However, it is also desirable to investigate a wider range of low-cost and abundant elements, such as Cu, Zn, Mo, and Sn, to either replace Ru or further increase the configurational entropy, which is anticipated to correspondingly enhance the electrothermal performance of the HEAs/LIG composite. (Page 5, Manuscript)

The senary FeCoNiCrMnRu sample also showed a slightly lower sheet resistance compared to both quinary FeCoNiCrRu ($\sim 60.2 \text{ } \Omega \text{ sq}^{-1}$) and FeCoNiCrMn ($\sim 59.9 \text{ } \Omega \text{ sq}^{-1}$) (Supplementary Figs. 37 and 38). (Page 9, Manuscript)

As for the high entropy samples, the FeCoNiCrMnRu showed a performance improvement in equilibrium temperatures at the 8V voltage compared to both FeCoNiCrRu and FeCoNiCrMn. Specifically, the equilibrium temperatures of the FeCoNiCrMnRu, FeCoNiCrRu, and FeCoNiCrMn were $\sim 204.1 \text{ } ^\circ\text{C}$, $\sim 183.4 \text{ } ^\circ\text{C}$, and $\sim 185.3 \text{ } ^\circ\text{C}$, respectively (Supplementary Figs. 50 and 51). (Page 12, Manuscript)

The infrared emissivity was consistently high across all the FeCoNiCrMnRu, FeCoNiCrRu, and FeCoNiCrMn samples, further indicating that this property is predominantly governed by the LIG carrier rather than the specific HEA composition. (Supplementary Fig. 56). (Page 13, Manuscript)

Reviewer's Comments

5. High-performance Joule heating applications of the FeCoNiCrMnRu HEA-NPs should consider the oxidation problems of the particles. It will be oxidized quickly in air.

Our Response

We sincerely appreciate the reviewer for raising the point regarding the oxidation stability of the FeCoNiCrMnRu HEA-NPs for Joule heating applications, which is indeed a common and significant concern for metallic nanoparticles. We fully agree with the reviewer that the oxidation of metallic NPs in air and during heating cycles can severely degrade their performance. The emissivity of mid-infrared band (2.5–20 μm) was measured by a Fourier transform infrared spectrometer with a gold integrating sphere (PIKE). To address this concern, we have conducted a thorough post-analysis of samples that were synthesized approximately six months ago and have since been exposed to ambient air (including periods of high humidity) and multiple Joule heating/cooling cycles. The performance of these aged HEAs/LIG samples remained remarkably stable, with negligible changes in both electrical resistance and emissivity (**Figure R29**). We further performed detailed TEM and EDS analysis on these aged samples. The results clearly demonstrate that the HEA-NPs maintained their metallic character without significant oxidation (**Figure R30**). The EDS analysis show an oxygen signal associated with the HEA-NPs (**Figure R31**). We attribute this exceptional oxidation resistance to the unique core-shell structure formed during our femtosecond laser processing. The HEA-NPs are predominantly encapsulated by a few layers of graphene. The graphene shell can serve as a protective layer, preventing the oxidation of the underlying metallic HEA cores during prolonged operation, thereby enhancing the long-term stability of the heater [*Adv. Mater.*, 2024, 36, 2402391]. Our work demonstrates that this strategy is equally effective for stabilizing

complex HEA systems in oxidizing atmospheres and under thermal conditions, which is a key advantage for the practical application of our material in durable, high-performance heating elements.

Figure R29. (a) Sheet resistance and (b) infrared emissivity of the HEAs/LIG in the wavelength range of 2.5 to 20 μm before and after stability test. A sample stored for eight months under ambient conditions with the temperature 20–25 $^{\circ}\text{C}$ and relative humidity of $\sim 70\%$, while the other subjected to extensive Joule heating-cooling cycles after 100 cycles.

Figure R30. HRTEM images of HEA-NPs after stability test, showing the multilayer graphene encapsulating the HEA-NPs.

Figure R31. HAADF-STEM images and the corresponding EDS elemental mappings for HEA-NPs (Fe, Co, Ni, Cr, Mn, Ru, and O) after stability test.

We have also added corresponding discussion about the stability of HEA-NPs in the revised manuscript. We also included **Figure R29** as **Figure S60**, **Figure R30** as **Figure S61**, and **Figure R31** as **Figure S62** in the supplementary information, as follows:

Furthermore, the HEAs/LIG samples also showed excellent durability and stability during the air exposure, high humidity, and repeated heating/cooling test. The sheet resistance and surface emissivity of both tested samples almost remained unchanged after stability test (Supplementary Fig. 60). This exceptional oxidation resistance was attributed to the unique core-shell structure formed during our femtosecond laser processing (Supplementary Fig. 61). The formed graphene shell encapsulating the HEA-NPs provides an excellent barrier against oxidation and moisture (Supplementary Fig. 62). The porous and robust LIG substrate firmly anchors the HEA-NPs, preventing their migration and agglomeration under repeated thermal treatment. (Page 13, Manuscript)

A List of changes in the main text of the revised manuscript

1. Added Figs. S60, S61, S62 and some discussions about the oxidation stability of the HEA-NPs in the main text (Page 13), according to Reviewers #1 and #3.
2. Added some discussions about the influence of the HEOs on the emissivity and conductivity in the main text (Page 8 and 10), according to Reviewer #1.
3. Modified the blackbody radiation curves in Fig. 4h, Figs. S54, and S58, according to Reviewer #1.
4. Added some uncertainty discussions and clear disclosure of material constants and boundary conditions in the supplementary information (Note S2 and S8), according to Reviewer #1.
5. Added Figs. S12, S13, S14 and some discussions about the large-area uniformity, batch-to-batch reproducibility, and practical throughput considerations of femtosecond direct writing in the main text (Page 5), according to Reviewer #1.
6. Added Fig. S53 and some discussions about the bending durability of HEAs/LIG in the main text (Page 12), according to Reviewer #1.
7. Added the detailed assumptions, parameters, and calculation approach in the supplementary information (Note S9), according to Reviewer #1.
8. Added some discussions about the material criticality and potential substitutions in the main text (Page 5), according to Reviewers #1 and #3.
9. Added Figs. S37, S38, S50, S51, S56 and some discussions in the main text (Page 9, 12, and 13) to show the role of Ru and configurational entropy, according to Reviewers #1 and #3.
10. Added Fig. S71 and some discussions in the main text (Page 15) to show the rapid on/off cycling stability of HEAs/LIG, according to Reviewer #1.
11. Added some discussions in the main text (Page 4) and modified the analysis in the supplementary information (Table S6) to show the practicality of using femtosecond laser for preparing HEA-NPs, according to Reviewer #2.
12. Added some discussions about the thermal simulation limitations in the main text (Page 4), according to Reviewer #2.
13. Added some discussions about the multifunctionality in the main text (Page 12), according to Reviewer #2.

14. Added Fig. S36 and some discussions in the main text (Page 8) to show the particle size of HEAs/LIG, according to Reviewer #3.

The Response to the Reviewer's Comments

Manuscript Number: NCOMMS-25-66444A

Title: Ultrafast achieving multiscale high-entropy alloys/graphene for high-performance Joule heating

Authors: Lingxiao Wang, Kai Yin, Jianqiang Xiao, Xinghao Song, Jiaqing Pei, Jun He, Ji-An Duan

We are indebted to the reviewers for their time and effort in reviewing our paper. The detailed comments from the referees are very impressive and greatly valued. We are happy to have such excellent advice for improving the paper's quality. With our best efforts and best knowledge, we have made changes by following the referees' comments. This report summarizes our understanding of the recommendations and the changes we have made in response. The reviewers' comments are reproduced following by the corresponding modifications to the manuscript.

In this report, some reference numbers, equations, and figures are marked by an R, for example, Figure R1, to distinguish them from those numbers for Figures in our original paper, for examples, Figure 1 and Figure S1. Different descriptions are also marked with different colors: Reviewer's comments are black; Our response is marked with blue and red; The part involved the revised manuscript in our response is red.

REVIEWER #1

Reviewer's Comments

The authors have addressed all concerns satisfactorily. Recommend acceptance for publication.

Our Response

Thank you for reviewing this work and supporting its publication in Nature Communications.

REVIEWER #2

Reviewer's Comments

The authors' revised discussion on femtosecond laser processing adopts a more conservative tone and demonstrates an acceptable academic depth compared with similar recent reports.

Our Response

We sincerely thank the reviewer for this positive assessment of the revised discussion on femtosecond laser processing and for recognizing the academic depth of our work.

Reviewer's Comments

1. The authors' response helps understand the rationale behind employing $E_T = 1/(h_c + h_r)$ as a normalized and comparative parameter for electrothermal performance. However, the terminology "energy-use efficiency" or "electrothermal conversion efficiency" used in the manuscript is then conceptually ambiguous. As Joule heating intrinsically converts electrical energy into heat with nearly 100% efficiency, E_T cannot be regarded as an energy-conversion efficiency in the thermodynamic sense.

Our Response

We sincerely thank the reviewer for this critical comment regarding the terminology "energy-use efficiency" or "electrothermal conversion efficiency". We agree that the Joule heating does indeed convert electrical energy to heat with nearly 100% efficiency. In our study, E_T was introduced as a performance index that reflects how effectively the generated heat is retained by the heater material under given convection and radiation conditions, rather than as a thermodynamic efficiency. In fact, many literature have also used this index to compare the heating efficiency of electric heaters [ACS Appl. Mater. Interfaces, 2012, 4, 2338-2342; Mater. Des., 2015, 86, 72-79; Carbon, 2019, 144, 116-126; Small, 2022, 18, 2202906]. To avoid conceptual ambiguity, we will revise the terminology throughout the manuscript. Accordingly,

the abbreviation of E_T (electrothermal conversion efficiency) has been changed to HEE (heater energy efficiency) in the following response.

In the revised manuscript, we have replaced the terminology “electrothermal conversion efficiency” with “heater energy efficiency” where appropriate, as follows:

Notably, the multiscale composite material (HEAs/LIG) can serve as Joule heating material, achieving a heater energy efficiency of $\sim 285.4 \text{ }^\circ\text{C cm}^2 \text{ W}^{-1}$. (Page 2, Manuscript)

We further established the relationship between equilibrium temperature and input power density, and summarized the heater energy efficiency of other advanced electrothermal materials. (Page 12, Manuscript)

As for the HEAs/LIG, the fitting calculation showed a heater energy efficiency of up to $\sim 285.4 \text{ }^\circ\text{C cm}^2 \text{ W}^{-1}$, indicating superior performance that surpassed most electrothermal materials (Fig. 4g and Supplementary Table 7)⁴⁹⁻⁵¹. (Page 12, Manuscript)

Remarkably, the resulting HEAs/LIG functions as an outstanding Joule heater, delivering a heater energy efficiency of $\sim 285.4 \text{ }^\circ\text{C cm}^2 \text{ W}^{-1}$ and average infrared emissivity of ~ 0.98 across a wide wavelength range of 2.5–20 μm . (Page 17, Manuscript)

Reviewer’s Comments

2. Regarding this point, the “energy saving” demonstration in Fig. 5(g–i) requires further clarification. Given that Joule heating converts electricity into heat at nearly 100% efficiency regardless of heater type, the generated heat should be transferred to the surroundings through radiation and convection. As acknowledged by the authors, E_T corresponds to the inverse of the effective heat-transfer coefficient. Thus, the local $\Delta T/P$ ratio merely indicates that the film has reduced heat loss to its immediate surroundings. In this study, however, the stated purpose of the heater is to warm the environment by transferring heat outward. Consequently, a heater with a high E_T value acts as a good thermal insulator, not necessarily as an effective space heater.

Our Response

We sincerely appreciate this important conceptual point. As for the Joule heating, the electrical energy is converted into heat with nearly 100% efficiency for any resistive heater. In this sense, there is no fundamental thermodynamic conversion efficiency difference between our HEAs/LIG heater and a conventional one. In Fig. 5(g–i), the temperature we track is the surface temperature of a titanium sheet placed inside the house model (Fig. 5h), which acts as a heated object. The “energy saving” specifically refers to the ability to reach a given target temperature of a representative heated object inside the house model using lower electrical power, under identical external boundary conditions. That is, we compare how much electrical power is required for different heaters to bring the same object to the same temperature. As the reviewer notes, a larger HEE indicates a larger temperature rise for a given input power and can be viewed as a higher effective thermal resistance between the heater and its surroundings. At the same time, the HEAs/LIG heater exhibits high infrared emissivity, which enhances radiative heat transfer toward the interior of the house model, where the titanium sheet is located. It indicates that, for a given input power, more of the heat from the HEAs/LIG heater is effectively directed toward the intended target, thereby exhibiting better energy efficiency ratio.

Furthermore, we contend that high HEE and high infrared emissivity are not fundamentally contradictory. Theoretically, an exceptionally high infrared emissivity may reduce HEE due to increased radiative heat loss. However, our HEAs/LIG heater simultaneously exhibits both high HEE and infrared emissivity, demonstrating its exceptional energy utilization efficiency and radiative heating capability.

Reviewer’s Comments

3. It is therefore essential to clarify whether the temperature reported in Fig. 5h represents the surface temperature of the heater or the air temperature inside the house model. Furthermore, while the study claims high emissivity for the HEA heater, high emissivity would increase h_r and consequently lower E_T . This consideration appears inconsistent with the high “electrothermal conversion efficiency” for the HEA heater.

Our Response

We thank the reviewer for pointing out this missing clarification. In Fig. 5h, the plotted temperature corresponds to the upper surface temperature of the heated Ti sheet placed inside the house model, rather than the bulk air temperature. The design of the experiment device is similar to Fig. 5a. The Ti sheet was mounted at the center of the interior space and directly exposed to the radiative heating from the HEAs/LIG heater. Its upper surface temperature was monitored by a temperature sensor attached to the sheet, and the recorded value is demonstrated in Fig. 5h.

We agree with reviewer that this is a valid point. From a fundamental heat-transfer perspective, an increase in infrared emissivity directly leads to a higher radiative heat-transfer coefficient (h_r). All else being equal, this would result in a larger total heat-transfer coefficient and therefore a lower value of HEE. This relationship is correctly noted. However, h_r is not the sole determinant of HEE. The calculated HEE value reflects the combined effect of both radiative and convective heat-loss pathways from the heater surface. The prepared HEAs/LIG composite features a complex and porous micro/nano-structured surface. Previous literature suggests that this morphology can be effective at suppressing air convection near the HEAs/LIG surface through stabilizing the air boundary layer and reducing effective air flow [*Int. J. Heat Mass Tran.*, 2024, 220, 124941; *Ceram. Int.*, 2024, 50, 36792-36799; *Interceram.-Int. Ceram. Rev.*, 2021, 70, 38-45]. Thus, there is no fundamental contradiction. The high emissivity ensures efficient radiative output once the heater is heated, while the surface morphology minimizes convective losses. This synergy allows the heater to achieve a high steady-state temperature with low input power and then to release that heat primarily as useful infrared radiation.

In the revised manuscript, we have corrected the corresponding illustration to clarify the measurement:

At 5 V voltage, the HEAs/LIG heater could lift the upper surface temperature of the heated object from ~ -7.5 °C to a human-comfortable level of ~ 24.0 °C, while CEH failed under the same conditions (Fig. 5h and Supplementary Fig. 73). (Page 15, Manuscript)

Fig. 5h, Temperature changes of the upper surface of heated object inside the house when radiated by HEAs/LIG, P-LIG, and CEH, respectively. (Page 16, Manuscript)

The laser-treated Ti substrate was used as the heated object in the radiative thermal transfer experiment (Fig. 5b and 5h). (Page 17, Manuscript)

Reviewer's Comments

4. If the authors are unable to provide a convincing rebuttal to the above concerns, it indicates that the original novelty and conceptual foundation of the work are not sufficiently persuasive.

Our Response

We sincerely thank the reviewer for this critical comment. The novelty and conceptual foundation of our work do not solely depend on the interpretation of the heater energy efficiency HEE. While the clarification regarding HEE is an important technical point, the core innovations of this study are multifaceted and robust.

In our work, we demonstrate the ultrafast femtosecond laser solid-phase synthesis of FeCoNiCrMnRu HEA-NPs on the LIG for the first time. Utilizing super-black LIG substrates and the ultrashort femtosecond laser pulse, this approach enables ultrafast heating and cooling rates. This facilitates the successful synthesis of disordered HEA-NPs.

Furthermore, we found that the HEA precursors significantly reduced the electrical resistance and enhanced the electrothermal conversion performance of LIG. Density functional theory calculations revealed that the resistance reduction originates from the deoxygenation and renovation of LIG induced by HEA-NPs, coupled with an increased density of states owing to HEA-NPs loading. The HEAs/LIG demonstrates excellent heater energy efficiency ($\sim 285.4 \text{ }^\circ\text{C cm}^2 \text{ W}^{-1}$), superior to most reported electrothermal materials.

Previous studies indicate that HEA-NPs usually exhibit high visible-light absorption, whereas high-entropy oxides demonstrate high infrared emissivity. Our designed HEAs/LIG composite overcomes the limitation of HEA-NPs by loading them onto the LIG surface, achieving ultrahigh broadband infrared emissivity. Meanwhile, the femtosecond laser reconstruction of LIG surface can generate rougher hierarchical micro/nanostructures with feature sizes exceeding infrared wavelength, enhancing the interaction between infrared light and substrates for further improving the infrared emissivity. Consequently, the fabricated

HEAs/LIG achieves an average infrared emissivity of ~ 0.98 across the 2.5–20 μm , which outperforms most high emissivity materials.

Finally, we demonstrate a series of applications utilizing the HEAs/LIG as a Joule heater, including radiative thermal transfer, electric warming, and rapid ice melting. For heating applications in cold winter environments, our HEAs/LIG heater achieves a $\sim 49.1\%$ energy saving compared to conventional electric heater (CEH), highlighting its significant potential for energy-efficient thermal management.

REVIEWER #3

Reviewer's Comments

I am satisfied with the revision. It can be accepted now.

Our Response

Thank you for reviewing this work and supporting its publication in Nature Communications.

A List of changes in the main text of the revised manuscript

1. Changed the terminology “electrothermal conversion efficiency” with “heater energy efficiency” in the main text (Page 2, 12 and 17), according to Reviewer #2.
2. Added illustration about the recorded temperature of the heated object in the main text (Page 15, 16 and 17), according to Reviewer #2.

The Response to the Reviewer's Comments

Manuscript Number: NCOMMS-25-66444B

Title: Femtosecond laser synthesis of multiscale high-entropy alloys/graphene composites for high-performance Joule heating

Authors: Lingxiao Wang, Kai Yin, Jianqiang Xiao, Xinghao Song, Jiaqing Pei, Jun He, Ji-An Duan

We are indebted to the reviewers for their time and effort in reviewing our paper. The detailed comments from the referees are very impressive and greatly valued. We are happy to have such excellent advice for improving the paper's quality. With our best efforts and best knowledge, we have made changes by following the referees' comments. This report summarizes our understanding of the recommendations and the changes we have made in response. The reviewers' comments are reproduced following by the corresponding modifications to the manuscript.

In this report, some reference numbers, equations, and figures are marked by an R, for example, Figure R1, to distinguish them from those numbers for Figures in our original paper, for examples, Figure 1 and Figure S1. Different descriptions are also marked with different colors: Reviewer's comments are black; **Our response is marked with blue and red**; **The part involved the revised manuscript in our response is red.**

REVIEWER #1

Reviewer's Comments

*After carefully reviewing the latest comments from Reviewer #2 together with the authors' rebuttal, I am inclined to support acceptance of the manuscript, provided that the conclusions are interpreted within clearly defined boundaries. In my assessment, the remaining disagreement is primarily related to the scope and wording of the interpretation rather than to the validity of the experimental data or the scientific soundness of the work. The study convincingly demonstrates a novel femtosecond-laser enabled route for fabricating high-entropy alloy nanoparticles on LIG and establishes their ultrahigh emissivity and effective radiative heat delivery to a nearby target object. When "heater energy efficiency (HEE)" is understood as a comparative performance metric, namely, the electrical power required to achieve a specified heating outcome under identical conditions, there is no fundamental physical inconsistency between ultrahigh emissivity and the reported heating performance. Indeed, the "heater energy efficiency" metric is a widely recognized and commonly adopted parameter for comparing the energy utilization efficiency of various electro-thermal conversion devices, such as *Adv. Funct. Mater.* 2023, 33, 2213357, *Adv. Eng. Mater.* 2022, 24, 2200368, and *Small*, 2022, 18, 2202906. The method for calculating the "heater energy efficiency" adopted in the manuscript is also consistent with those reported in other literature.*

Regarding the "energy saving" and heat-transfer interpretation, I agree that the manuscript should not be read as demonstrating improved general space-heating efficiency. However, the experimental data robustly support a more focused and well-defined conclusion: the proposed radiative heating approach achieves measurable reductions in electrical power consumption required to reach the same target-object temperature, within the specific geometric and enclosed conditions tested. The authors have explicitly confined the "energy saving" claim to the aforementioned conditions. The energy-saving performance of the HEAs/LIG composites has been experimentally verified, and its significant energy-saving potential has further been validated through simulations. Additionally, the authors reasonably suggest that the LIG morphology contributes to suppressing convection, thereby maintaining

excellent “heater energy efficiency” even under conditions of high infrared emissivity, which should be regarded as a plausible contributing factor. Considering the above points collectively, I believe the manuscript is scientifically robust. With such restrained interpretation and minor textual refinement, the work makes a meaningful contribution to radiative heating materials and can be suitable for publication.

Our Response

Thank you for reviewing this work and supporting its publication in Nature Communications. We sincerely thank Reviewer #1 for the thorough assessment and strong support for publication. We have carefully addressed the recommendation for "minor textual refinement" and ensured our conclusions are "interpreted within clearly defined boundaries."

The specific revisions made in the manuscript are as follows:

Heater energy efficiency can be interpreted as a comparative performance metric, representing the electrical power density required to achieve a specified surface temperature under identical input conditions. (Page 10, Manuscript)

The high heater efficiency of HEAs/LIG represents an objective outcome derived after accounting for both infrared radiation and convective heat transfer. The higher heater efficiency of HEAs/LIG may also be plausibly attributed to the disruption of air convection by their surface structural morphology. (Page 10, Manuscript)

Due to the high heater energy efficiency and infrared emissivity, the HEAs/LIG could achieve a higher surface temperature under the same electrical power input. This elevated temperature, coupled with the high emissivity, enables efficient radiative heating of objects. (Page 12, Manuscript)

Under the tested specific geometric and enclosed conditions (housing model), HEAs/LIG heaters achieved a measurable reduction in the electrical energy consumption required to reach the same target object temperature, thereby demonstrating an energy-saving effect. (Page 12, Manuscript)

REVIEWER #2

Reviewer's Comments

This reviewer appreciates the multidisciplinary nature of the study and its contributions to femtosecond-laser synthesis of high-entropy alloy nanoparticles and broadband emissivity control. However, the claimed coexistence of ultrahigh emissivity and high heater energy efficiency (HEE) remains supported only qualitatively, and the demonstrated “energy saving” should be interpreted strictly in the context of radiative heating of a target object rather than as an improvement in general space-heating efficiency.

In particular, the assertion that the LIG surface morphology suppresses convection sufficiently to compensate for increased radiative losses is not convincingly validated. This is a nontrivial heat-transfer argument, and the cited literature is not directly related to this problem. Moreover, within a small “house model” enclosure, buoyancy-driven natural convection is difficult to suppress in practice. As such, the concept of an LIG-based space heater remains scientifically fragile and open to further debate.

Our Response

We sincerely thank Reviewer #2 for the continued engagement and acknowledgment of our contributions to femtosecond-laser synthesis and emissivity control. We have clarified that the "energy-saving effect" discussed in the paper specifically refers to the reduction in electrical energy consumption in the context of heating of a target object. This clarification has been consistently incorporated in the main text. In addition, a discussion has been incorporated regarding the possibility that the surface morphology enhances heater efficiency by disrupting thermal convection.

The specific revisions made in the manuscript are as follows:

Heater energy efficiency can be interpreted as a comparative performance metric, representing the electrical power density required to achieve a specified surface temperature under identical input conditions. (Page 10, Manuscript)

The high heater efficiency of HEAs/LIG represents an objective outcome derived after accounting for both infrared radiation and convective heat transfer. The higher heater efficiency of HEAs/LIG may also be plausibly attributed to the disruption of air convection by their surface structural morphology. (Page 10, Manuscript)

Due to the high heater energy efficiency and infrared emissivity, the HEAs/LIG could achieve a higher surface temperature under the same electrical power input. This elevated temperature, coupled with the high emissivity, enables efficient radiative heating of objects. (Page 12, Manuscript)

Under the tested specific geometric and enclosed conditions (housing model), HEAs/LIG heaters achieved a measurable reduction in the electrical energy consumption required to reach the same target object temperature, thereby demonstrating an energy-saving effect. (Page 12, Manuscript)

A List of changes in the main text of the revised manuscript

1. Tempered and clarified the claims related to LIG morphology, heater energy efficiency, and energy saving in the main text (Page 10 and 12), according to Reviewers #1 and #2.